# Federated Manifold Learning (FML): Tackling Domain Heterogeneity with Structural Knowledge Transfer

Xutong Mu [1]  Yanbiao Ma [2 3 4]  Jia Shi [5]  Xueli Geng [5]  Fengkai Xiang [1]  Tao Zhang [1]  Ke Cheng [1]  Yulong Shen [1]

## Abstract

Federated Learning (FL) faces significant challenges due to domain heterogeneity, where data from different clients exhibit substantial statistical shifts that hinder the generalization of the global model. Although existing methods attempt to mitigate this by exchanging class prototypes, they fall short by representing an entire class's complex distribution with a single point. This oversimplification disregards the rich structural information within the data, especially across diverse domains. To address this limitation, we propose a paradigm shift from point-based representation to structure-based knowledge transfer. We introduce Federated Manifold Learning (FML), a novel framework that leverages perceptual manifolds—the intrinsic geometric structures of classes in the feature space—as rich knowledge carriers. In FML, clients transmit compressed manifolds, which are adaptively fused on the server using an attention-based Manifold Mutual Learning (MML) mechanism. This process enables domain-specific structures to learn from each other, creating a unified yet flexible global convergence target. Manifold-guided local training, enforced by a manifold approximation loss and a separation loss, further aligns local models with this global structure. Extensive experiments on the Digits and Office31 benchmarks demonstrate that FML substantially outperforms state-of-the-art methods, achieving accuracy improvements of up to 6.47%.

[1]School of Computer Science and Technology, Xidian University, Xi'an, China [2]Gaoling School of Artificial Intelligence, Renmin University of China, Beijing, China [3]Beijing Key Laboratory of Research on Large Models and Intelligent Governance, Beijing, China [4]Engineering Research Center of Next-Generation Intelligent Search and Recommendation, MOE, Beijing, China [5]School of Artificial Intelligence, Xidian University, Xi'an, China. Correspondence to: Yanbiao Ma <ybma1998@ruc.edu.cn>.

*Proceedings of the 43$^{rd}$ International Conference on Machine Learning*, Seoul, South Korea. PMLR 306, 2026. Copyright 2026 by the author(s).

## 1. Introduction

Federated Learning (FL) represents a paradigm focused on privacy protection (McMahan et al., 2017; Yang et al., 2019; Mu et al., 2024b; Lao et al., 2023), enabling collective model training without compromising data confidentiality (Li et al., 2024a; Qi et al., 2023; Meng et al., 2024; Qi et al., 2025b). While effective on homogeneous data, its performance is notoriously compromised when facing data heterogeneity—a common scenario in the real world (Tan et al., 2023; Li et al., 2020a; 2026c;a). Much of the existing research has focused on *label heterogeneity*, where clients have different distributions of classes (Ye et al., 2023; Meng et al., 2024; Qi et al., 2025b;a). However, a more challenging and realistic problem is **domain heterogeneity**: data from different clients not only vary in labels but also originate from entirely distinct visual domains (Ye et al., 2023; Huang et al., 2023b; 2022; 2023a).

Consider a practical task of digit recognition: one client's data might be clean, grayscale MNIST images, while another's could be colorful digits from complex real-world scenes like SVHN (LeCun et al., 1998; Netzer et al., 2011). Although both datasets represent the same classes (digits '0' through '9'), their underlying feature distributions are vastly different. A model optimized for MNIST's clean style will inevitably struggle with SVHN's clutter, and vice-versa. Simply averaging the parameters from these specialist models, as done in FedAvg, often results in a global model that is mediocre across all domains, failing to generalize effectively (Huang et al., 2023b; 2022; Hu et al., 2024; Wu et al., 2026). This challenge—learning a single, powerful global model from disparate domains—is the central focus of our work.

Recent advanced methods have attempted to tackle heterogeneity by moving beyond parameter averaging, proposing the exchange of higher-level knowledge signals, such as class prototypes (Tan et al., 2022; Mu et al., 2023; Long et al., 2023; Huang et al., 2023b). These methods compute a representative feature vector (a prototype) for each class on each client, which are then aggregated on the server to form global prototypes. These global signals then guide local training, aiming for a more consistent feature space.

However, these prototype-based approaches suffer from a fundamental limitation: **they compress the entire, rich distribution of a class into a single point in the feature space.** This oversimplification is particularly detrimental under domain heterogeneity. The prototype of an MNIST '7' (a point in a "clean" feature region) and an SVHN '7' (a point in a "cluttered" feature region) are inherently different. Simply averaging them creates a global prototype that accurately represents *neither* domain and discards the crucial information about how each domain's data is structured. A single point is simply insufficient to capture the complexity and diversity of a class as it appears across multiple domains.

Recognizing this limitation, we advocate for a paradigm shift: from representing a class as a **point** to capturing its intrinsic geometric **structure**. We posit that features of the same class, even across different domains, lie on or near a low-dimensional **Perceptual Manifold** within the high-dimensional feature space (Lei et al., 2020; Tenenbaum et al., 2000; Li et al., 2024b; 2025a; Liao et al., 2025). This manifold, akin to a class's 'footprint' or 'constellation', naturally encodes the intra-class variations and diversity that single prototypes discard. By sharing and fusing these rich structural representations, we can create a unified convergence target that respects, rather than erases, domain-specific characteristics (Ma et al., 2022; 2023a;b; Li et al., 2025b).

We introduce **Federated Manifold Learning (FML)**, a novel framework where clients transmit compressed representations of perceptual manifolds instead of model parameters or single prototypes. On the server, the **Manifold Mutual Learning (MML)** mechanism utilizes attention to effectively fuse cross-domain knowledge by enabling manifolds to selectively learn from each other. The optimized global manifolds subsequently guide local training through a manifold approximation loss for improved generalization and a separation loss for enhanced discrimination. Our key contributions are summarized as follows:

- We systematically investigate domain heterogeneity in Federated Learning (FL) and identify the fundamental limitation of existing prototype-based methods in effectively representing, generalizing, and adapting across cross-domain classes. Our comprehensive analysis highlights the inherent challenges faced by these methods in capturing the diverse and complex characteristics of classes across different domains, especially in dynamic, real-world environments.

- We propose Federated Manifold Learning (FML), a new paradigm that leverages perceptual manifolds as knowledge carriers. FML addresses heterogeneity by establishing a consistent yet flexible convergence target, significantly enhancing generalization.

- We introduce key components for Federated Manifold Learning (FML), including Perceptual Manifold Compression (PMC), an attention-based Manifold Mutual Learning (MML) mechanism, and tailored loss functions, which collectively enable efficient, scalable, and robust manifold-driven training with significantly enhanced performance, generalization, and adaptability across diverse and dynamic domains.

- Extensive experiments on standard domain heterogeneity benchmarks (Digits (Hull, 1994; LeCun et al., 1998; Netzer et al., 2011; Wang et al., 2015) and Office31 (Saenko et al., 2010) ) demonstrate that FML substantially outperforms state-of-the-art methods, with accuracy improvements of up to **6.47%**.

**Conflict of Interest Disclosure.** The authors have no financial conflicts of interest to disclose.

## 2. Related Work

To address the data heterogeneity challenge in federated learning, researchers introduce various constraints on parameter updates, preventing client models from deviating too much from the shared global model during local learning. FedProx (Li et al., 2020b) employs an $\ell_2$-norm distance to constrain the distance between the local and global models, thereby ensuring stability. Scaffold (Karimireddy et al., 2020) applies control variables at the central server and among participants to correct client drift, facilitating better coordination. FedDC (Gao et al., 2022) compensates for differences in client updates by introducing learned local drift variables, imposing consistency constraints at the parameter level, thereby improving model convergence. FedDyn (Acar et al., 2021) optimizes local objectives dynamically, achieving progressive consistency between local optima and global objectives, which accelerates training convergence. FedMA (Wang et al., 2020) constructs a shared global model hierarchically by matching and averaging weights, fostering greater global alignment. MOON (Li et al., 2021) uses contrastive learning (Chen et al., 2020) to constrain local training, enabling local models to learn representations that are closer to the global model and superior to those of the previous round, thus improving overall performance.

In recent years, prototype learning (Snell et al., 2017), as an effective method for knowledge sharing and transfer, has seen significant development in the field of federated learning. Researchers explore promoting knowledge transfer between clients through prototype exchange. For instance, FedProto (Tan et al., 2022) proposes an innovative strategy that reduces communication costs by exchanging prototype representations of categories between clients and servers, rather than traditional gradients or model parameters. FedProc (Mu et al., 2023) introduces the concept of a global

prototype, serving as a reference point to enhance client training quality during local update processes. Additionally, FedProc encourages enhancing intra-class feature compactness and inter-class feature distinctiveness by applying a contrast loss function. FedCD (Long et al., 2023) introduces fine-grained prototypes to rebalance the class distribution on each client and correct the classification layer of each local model. FedCSPC (Qi et al., 2023) calibrates data prototypes by employing clustering to learn data patterns, then enhances the robustness of this calibration using an augmented contrastive learning approach. However, these methods assume that all data belong to the same domain and simulate the diversity of real-world data through variations in the number of classes across different clients.

Federated learning faces not only quantitative differences but also significant disparities in data features, meaning data may come from different domains (Ye et al., 2023; Li et al., 2026b; 2025c; 2024c). FCCL+ (Huang et al., 2023a) tackles domain heterogeneity by aligning instance similarity distributions at logical and feature levels with cross-correlation matrices, enhancing communication efficiency and generalization. However, it focuses more on improving personalized models than a shared global model. The FPL (Huang et al., 2023b) scheme revisits prototypes, defining cluster prototypes and unbiased prototypes. It aligns local instances with these prototypes using consistency regularization techniques. Nonetheless, FPL uses only one prototype to represent a category or a domain's data, which cannot fully capture the complexity of the data distribution. Fed3R (Fanì et al., 2024) uses pre-trained feature extractors and computes a Ridge Regression classifier in closed form. However, its primary limitation lies in its reliance on linear assumptions for classification, which may not fully capture the complexities of non-linear data distributions. FedDIM (Liao et al., 2025) captures fine-grained decision-making processes from feature vectors and classifier weights, promoting invariance through a regularization term. However, its effectiveness is dependent on the consistency of the domain distributions across clients. FedDP (Fu et al., 2025) uses domain-independent prototypes to address feature shift issues by aligning the representation and parameter spaces across clients. However, the global model's performance is suboptimal in certain domains, highlighting a limitation in its ability to generalize across all domains. FedGWC (Leo et al., 2025) clusters clients with similar data distributions. This enables personalized model training for each group, enhancing performance in heterogeneous environments. However, a key limitation of the approach is the computational overhead associated with clustering.

**Manifold-based Federated Learning.** Several recent studies have explored manifold modeling to address client data heterogeneity in federated learning. A representative line of work uses manifolds as regularizers or calibration tools in feature space. For example, FedMRUR (An et al., 2023) introduces a hyperbolic graph manifold regularizer to reduce local-global inconsistency, FedMR (Fan et al., 2023) reshapes the feature manifold under partially class-disjoint data, and FedMC (Ma et al., 2026) models nonlinear local geometry and calibrates local features with a global geometry dictionary. Another related direction focuses on manifold alignment, such as FedMP (Zhou et al., 2025), which performs stochastic feature manifold completion and aligns client manifolds with class prototypes. Different from these methods, FML treats manifolds as explicit and communication-efficient knowledge carriers: clients compress local class manifolds into representative points, the server adaptively fuses them through attention-based manifold mutual learning, and the resulting global manifold is fed back to guide local training.

**Robust and Secure Federated Learning.** Orthogonal to statistical and domain heterogeneity, federated learning is also vulnerable to malicious clients, poisoning attacks, and Byzantine failures. Several studies have investigated robust client detection and aggregation mechanisms to mitigate these security threats. For example, FedDMC detects malicious clients to improve the robustness and efficiency of federated learning, while FedPTA formulates malicious-client detection through prior-based tensor approximation (Mu et al., 2024b;a). FedRMA further considers robustness against multiple poisoning attacks, and FedCWA performs credibility-weighted aggregation for Byzantine-robust federated learning (Xiao et al., 2024; Musa et al., 2025). These methods defend federated optimization against adversarial participants, while FML focuses on cross-domain structural knowledge transfer under domain heterogeneity, making them complementary to our work. Beyond training-time robustness, secure and lightweight inference has also been explored to protect deployed neural networks in resource-constrained environments (Song et al., 2024).

## 3. Federated Manifold Learning

The Federated Manifold Learning (FML) framework effectively addresses domain heterogeneity by leveraging perceptual manifolds for efficient and robust knowledge transfer across diverse and dynamic domains. As illustrated in Figure 1, the process consists of three key stages: 1) **Client-Side Extraction**: Clients extract compressed perceptual manifolds from local data and upload them, along with model parameters, to the central server, ensuring efficient data transmission. 2) **Server-Side Fusion**: The server aggregates these manifolds from all clients and orchestrates mutual learning to generate enhanced, domain-aware global manifolds that effectively capture shared knowledge and facilitate better collaboration. 3) **Client-Side Guided Train-**

**ing**: Clients then download the global manifolds and use them to guide local training, ensuring that their feature spaces align with the global structure for improved model generalization and enhanced performance on unseen data.

We consider a federated system with $M$ clients, where client $m$ holds a private dataset $\mathcal{D}_m = \{(x_i, y_i)\}_{i=1}^{N_m}$. Due to domain heterogeneity, feature distributions differ significantly across clients despite sharing a label space. The model comprises a feature extractor $f(w_e; \cdot)$ and a classifier $g(w_c; \cdot)$. For input $x$, the extractor outputs a feature $z = f(w_e; x) \in \mathbb{R}^d$, which the classifier maps to logits $s = g(w_c; z) \in \mathbb{R}^{|I|}$. Our goal is to train a global model $w = \{w_e, w_c\}$ that generalizes across all domains.

### 3.1. Perceptual Manifold Compression

For each class $l$, client $m$ extracts feature vectors $Z_{m,l}$ using $f_m$, forming a perceptual manifold in $\mathbb{R}^d$. Transmitting the full $Z_{m,l}$ is communication-prohibitive due to its size.

**Motivation.** We aim to compress $Z_{m,l}$ into a small set of $r$ representative points, $Z'_{m,l}$, while preserving its intrinsic structural diversity, quantified by the manifold's volume. The volume of $Z = [z_1, \ldots, z_n]$ measures the spanned space and information content. It is computed via the determinant of the covariance matrix regularized by an identity matrix $I$ (Ma et al., 2023a):

$$\text{Vol}(Z) = \frac{1}{2} \log_2 \det(I + \frac{1}{n}(Z - \bar{Z})(Z - \bar{Z})^T), \quad (1)$$

where $\bar{Z}$ is the mean of the feature set $Z$.

We introduce the Information Efficiency Index (IEI) to balance compression and information preservation:

$$IEI = \frac{v/v_{\max} + 1}{r/n + 1}, \quad (2)$$

where $v$ and $v_{\max}$ are the volumes of the compressed ($r$ points) and original ($n$ points) manifolds, respectively. Maximizing IEI yields high information content with minimal cost. To achieve this, we employ Fast Greedy Determinantal Point Process (Fast Greedy DPP) selection (Chen et al., 2018), which efficiently chooses geometrically diverse representative points. Compared with exact DPP inference (Kulesza et al., 2012), Fast Greedy DPP avoids expensive combinatorial search by greedily maximizing the marginal determinant gain. The full Perceptual Manifold Compression (PMC) process, which iteratively selects the optimal $r$ maximizing IEI via Fast Greedy DPP, is detailed in Algorithm 1.

### 3.2. Manifold Mutual Learning

Upon receiving the compressed manifolds $\{Z'_{m,l}\}_{m=1}^M$ for each class $l$ from all clients, the server aggregates them into a global manifold $\hat{Z}_l$.

---

**Algorithm 1** Perceptual Manifold Compression (PMC)

1: **Input:** Feature set $Z = [z_1, \ldots, z_n] \in \mathbb{R}^d$
2: **Output:** Compressed manifold $Z'$
3: Initialize:
4: $r \leftarrow 2, IEI_{\text{prev}} \leftarrow 0, Z' \leftarrow \emptyset, v_{\max} \leftarrow \text{Vol}(Z)$
5: **while** true **do**
6:     $Z'_{\text{temp}} \leftarrow \text{Fast Greedy DPP}(Z, r)$
7:     $v \leftarrow \text{Vol}(Z'_{\text{temp}})$
8:     $IEI \leftarrow \dfrac{v/v_{\max} + 1}{r/n + 1}$
9:     **if** $IEI > IEI_{\text{prev}}$ **then**
10:         $IEI_{\text{prev}} \leftarrow IEI; Z' \leftarrow Z'_{\text{temp}}$
11:     **else**
12:         **break**       ▷ max IEI reached
13:     **end if**
14:     $r \leftarrow r + 1$
15: **end while**
16: **return** $Z'$

---

**Motivation.** A naive averaging of these manifolds would wash out domain-specific features, repeating the mistake of prototype-based methods at a structural level. To enable a more nuanced and effective fusion of knowledge, we propose a Manifold Mutual Learning (MML) mechanism. The core idea is to first identify distinct and meaningful data patterns within the aggregated manifold and then allow these patterns to selectively learn from each other, thereby enhancing the overall knowledge transfer and improving model robustness across diverse domains.

First, we apply the parameter-free FINCH clustering algorithm (Sarfraz et al., 2019) to the global manifold $\hat{Z}_l$. This partitions the manifold into several clusters $\{T_{l,s}\}$, each representing a distinct geometric structure, likely corresponding to a specific domain or a mixture of similar domains.

Next, to facilitate knowledge exchange between these clusters, we employ an attention mechanism (Vaswani et al., 2017). For each target cluster $T_{l,s}$ (acting as the Query), we allow it to attend to all other clusters (acting as Keys and Values). This process computes a weighted sum of features from other clusters, generating an updated, context-aware representation $T'_{l,s}$ that has absorbed relevant information from its peers. The attention function is defined as:

$$T'_{l,s} = \text{ApplyAttention}(Q, K, V) = \text{Softmax}\left(\frac{QK^T}{\sqrt{d_k}}\right)V. \quad (3)$$

The original cluster representation is then updated via a weighted sum with the new representation, controlled by a hyperparameter $\gamma$. This process ensures that domain-specific knowledge is enriched, integrated, and seamlessly incorporated, not replaced, thus preserving the diversity of information across different domains while enhancing the overall model performance. Finally, the server reconstructs

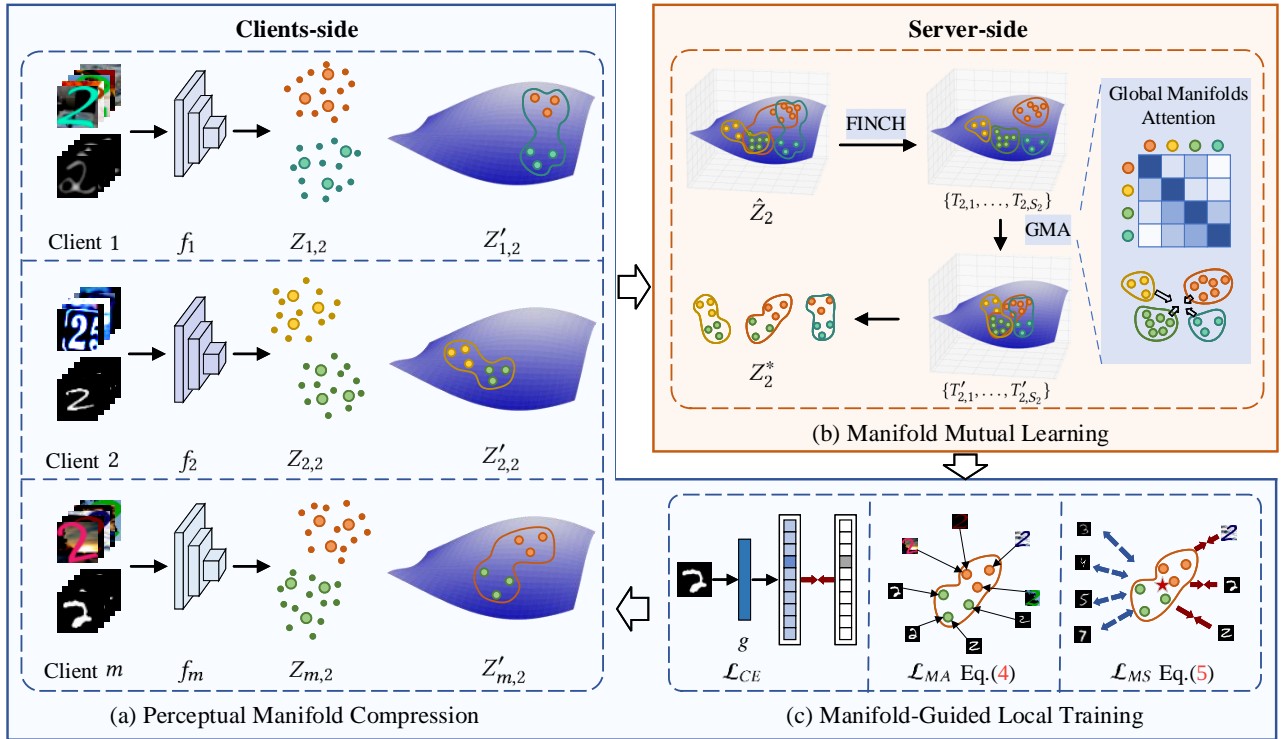

*Figure 1.* Architecture illustration of Federated Manifold Learning (FML). This consists of three key phases: (a) Perceptual Manifold Compression §3.1; (b) Manifolds Mutual Learning §3.2; (c) Manifold-Guided Local Training §3.3.

the enhanced global manifolds $\{Z_{m,l}^*\}$ for each client and distributes them accordingly. The detailed MML process is thoroughly outlined in Algorithm 2.

**Scalability of MML.** It is worth noting that MML operates on highly compressed manifold points rather than raw local features. For each class $l$, each client uploads only $r$ representative points after PMC, so the input size of FINCH is at most $M \times r$, where $M$ is the number of clients. FINCH then partitions these compressed points into $S_l$ structure-driven clusters, whose number is determined by the feature geometry rather than directly fixed by the number of clients. The subsequent attention module is applied to cluster-level representations, with a cost of $O(S_l^2 d_k)$, instead of being performed over all raw features. Since $r$ is small in our implementation, the server-side computation of MML remains practical.

### 3.3. Manifold-Guided Local Training

After receiving the global model and the enhanced manifolds $Z_l^*$ from the server, each client proceeds with local training. The training is guided by a composite loss function designed to align the local feature space with the global manifold structure. The total loss is $\mathcal{L} = \mathcal{L}_{CE} + \alpha \mathcal{L}_{MA} + \beta \mathcal{L}_{MS}$.

The first component is the standard Cross-Entropy Loss ($\mathcal{L}_{CE}$) for classification (Mao et al., 2023). The other two components are manifold-guided regularizers:

**Manifold Approximation Loss (MAloss).** This loss serves as an attractive force, pulling local feature representations towards the global manifold structure. For each local sample $x_i$ with feature $z_i$, we find its closest anchor point $z_j^*$ on the downloaded manifold $Z_{y_i}^*$ (where $y_i$ is the label of $x_i$). The loss minimizes the distance between them:

$$\mathcal{L}_{MA} = \sum_{i=1}^{n} \left\| z_i - z_{j(i)}^* \right\|_2^2,$$
$$j(i) = \underset{j:\, z_j^* \in Z_{y_i}^*}{\arg\min} \left\| z_i - z_j^* \right\|_2^2. \tag{4}$$

Here, $j(i)$ denotes the index of the closest anchor point on the downloaded class manifold $Z_{y_i}^*$ for sample $x_i$. This encourages the local model to produce features that lie on or near the rich geometric structure learned from all domains, thereby ensuring better alignment with the global manifold and improving the model's generalization ability across diverse tasks and environments.

**Manifold Separation Loss (MSloss).** While MAloss ensures intra-class compactness around the manifold, this loss provides a repulsive force to ensure inter-class separability. Using principles of contrastive learning, it pushes the

**Algorithm 2** Manifold Mutual Learning (MML)

1: **Input:** Manifolds from $M$ clients $\{\mathcal{Z}_m\}_{m=1}^{M}$
2: **Output:** Learned manifolds for $M$ clients $\{\mathcal{Z}_m^*\}_{m=1}^{M}$
3: **for** $l = 1$ **to** $c$ **do**
4:    $\hat{Z}_l \leftarrow [\,]$                              ▷ concat buffer
5:    **for** $m = 1$ **to** $M$ **do**
6:       $\hat{Z}_l \leftarrow \text{Concat}(\hat{Z}_l, Z'_{m,l})$
7:    **end for**
8:    $\{T_{l,s}\}_{s=1}^{S_l} \leftarrow \text{FINCH}(\hat{Z}_l)$
9:    **for** $s = 1$ **to** $S_l$ **do**
10:       $Q \leftarrow \text{ExtractQuery}(T_{l,s})$
11:       $K \leftarrow [\,]; \quad V \leftarrow [\,]$
12:       **for** $s' = 1$ **to** $S_l$ **do**
13:          **if** $s' \neq s$ **then**
14:             $K \leftarrow \text{Concat}(K, \text{ExtractKey}(T_{l,s'}))$
15:             $V \leftarrow \text{Concat}(V, \text{ExtractValue}(T_{l,s'}))$
16:          **end if**
17:       **end for**
18:       $T'_{l,s} \leftarrow \text{ApplyAttention}(Q, K, V)$
19:       $T_{l,s} \leftarrow (1 - \gamma)\, T_{l,s} + \gamma\, T'_{l,s}$
20:    **end for**
21:    **for** $m = 1$ **to** $M$ **do**
22:       $Z_{m,l}^* \leftarrow \text{Reconstruct}(\{T_{l,s}\}_{s=1}^{S_l})$
23:    **end for**
24: **end for**
25: **for** $m = 1$ **to** $M$ **do**
26:    $\mathcal{Z}_m^* \leftarrow \{Z_{m,1}^*, Z_{m,2}^*, \ldots, Z_{m,c}^*\}$
27: **end for**
28: **return** $\{\mathcal{Z}_m^*\}_{m=1}^{M}$

---

feature vector $z_i$ of a given class $l$ closer to its own class's manifold centroid $\bar{Z}_l^*$, while simultaneously pushing it away from the centroids of all other class manifolds $\bar{Z}_{l'}^*$:

$$\mathcal{L}_{MS} = -\log \frac{\exp(\text{sim}(z_i, \bar{Z}_l^*))}{\sum_{k \in I} \exp(\text{sim}(z_i, \bar{Z}_k^*))}, \qquad (5)$$

where $\text{sim}(\cdot, \cdot)$ denotes cosine similarity. This enhances the model's discriminative power by enforcing a clear separation between the learned manifold structures of different classes. The overall federated manifold learning algorithm is shown in Algorithm 3.

### 3.4. Convergence Analysis

We provide a mechanism-aware convergence analysis for FML. The server-side update is written as $w^{t+1} = w^t - \eta d_t$, where $d_t$ denotes the effective update direction induced by local training, manifold compression, and attention-based fusion. Let

$$\|d_t - \nabla\mathcal{F}(w^t)\| \leq \delta \|\nabla\mathcal{F}(w^t)\|, \quad \delta = \delta_{\text{comp}} + \delta_{\text{att}} < 1.$$

Here, $\delta_{\text{comp}}$ and $\delta_{\text{att}}$ respectively measure the distortion from manifold compression and the bias from attention-based fusion.

**Theorem 3.1** (Mechanism-Aware Linear Convergence of FML). *Suppose $\mathcal{F}(w)$ is $L$-smooth and satisfies a local PL*

---

**Algorithm 3** The FML Framework

1: **Input:** Datasets $\{\mathcal{D}_m\}_{m=1}^{M}$, rounds $T$, clients $M$, epochs $E$, classes $K$, lr $\eta$.
2: **Output:** Final global model $w^T$.
3: **Server Process:**
4: Initialize $w^0$, $\{\mathcal{Z}_m^*\}_{m=1}^{M}$.
5: **for** $t = 0$ **to** $T - 1$ **do**
6:    **for** $m = 1$ **to** $M$ **in parallel do**
7:       Send $w^t$ and $\mathcal{Z}_m^*$ to client $m$.
8:       $w_m^{t+1}, \mathcal{Z}_m'^{t+1} \leftarrow \text{LocalTraining}(m, w^t, \mathcal{Z}_m^*)$
9:    **end for**
10:    $w^{t+1} \leftarrow \sum_{m=1}^{M} \frac{|\mathcal{D}_m|}{|\mathcal{D}|} w_m^{t+1}$           ▷ **FedAvg**
11:    $\{\mathcal{Z}_m^*\}_{m=1}^{M} \leftarrow \text{MML}(\{\mathcal{Z}_m'^{t+1}\}_{m=1}^{M})$
12: **end for**
13: **return** $w^T$

14: **Function** LocalTraining$(m, w, \mathcal{Z}^*)$:
15: Split $w$ into encoder $w_e$ and classifier $w_c$.
16: **for** epoch $= 1$ **to** $E$ **do**
17:    **for** each batch $(x_i, y_i) \in \mathcal{D}_m$ **do**
18:       $z_i \leftarrow f(w_e; x_i)$
19:       $L_{CE} \leftarrow -\sum_i y_i \log(g(w_c; z_i))$
20:       $L_{MA} \leftarrow \sum_i \left\| z_i - z_{j(i)}^* \right\|^2$
21:       $L_{MS} \leftarrow -\log \frac{\exp(\text{sim}(z_i, \bar{Z}_{y_i}^*))}{\sum_{l'} \exp(\text{sim}(z_i, \bar{Z}_{l'}^*))}$
22:       $L \leftarrow L_{CE} + \alpha L_{MA} + \beta L_{MS}$
23:       Update $w \leftarrow w - \eta\nabla L$
24:    **end for**
25: **end for**
26: **for** $l = 1$ **to** $K$ **do**
27:    $\mathcal{Z}_{m,l}' \leftarrow \text{PMC}(\{f(w_e; x) \mid (x, y) \in \mathcal{D}_m, y = l\})$
28: **end for**
29: **return** $w, \{\mathcal{Z}_{m,1}', \ldots, \mathcal{Z}_{m,K}'\}$

---

*condition with constant $\mu > 0$. If*

$$0 < \eta < \frac{2(1 - \delta)}{L(1 + \delta)^2},$$

*then for any $T \geq 1$,*

$$\mathcal{F}(w^T) - \mathcal{F}^\star \leq \rho^T \left[ \mathcal{F}(w^0) - \mathcal{F}^\star \right],$$

*where*

$$\rho = 1 - 2\mu\eta \left[ (1 - \delta) - \frac{L\eta}{2}(1 + \delta)^2 \right] < 1.$$

This result shows that FML preserves linear convergence under bounded mechanism-induced deviation. When $\delta_{\text{comp}} = \delta_{\text{att}} = 0$, it reduces to the idealized gradient-descent case. The full proof is provided in the Appendix.

## 4. Experiments

### 4.1. Experimental Setup

**Datasets and Model.** The performance of FML is evaluated using two standard benchmark datasets: Digits (Hull, 1994; LeCun et al., 1998; Netzer et al., 2011; Wang et al., 2015) and Office31 (Saenko et al., 2010).

*Table 1.* Comparison with the SOTA methods on Digits and Office31 tasks. AVG denotes the weighted average accuracy calculated for all domains. Best in bold and second in underline. Please see details in Sec. 4.2.

| Methods | Digits | | | | | | Office31 | | | | |
|---|---|---|---|---|---|---|---|---|---|---|---|
| | MNIST | SVHN | SYN | USPS | AVG | Δ | Amazon | Webcam | DSLR | AVG | Δ |
| FedAvg (McMahan et al., 2017) | 93.32 | 41.24 | 76.64 | 84.63 | 79.44 | — | 48.78 | 58.60 | 47.07 | 50.48 | — |
| FedProx (Li et al., 2020b) | 93.46 | 40.26 | 72.98 | 85.20 | 77.07 | -2.36 | 48.23 | 68.81 | 44.67 | 51.80 | 1.32 |
| MOON (Li et al., 2021) | 94.94 | 31.19 | 65.28 | 83.56 | 71.90 | -7.54 | 41.51 | 36.27 | 40.23 | 40.34 | -10.14 |
| FedProto (Tan et al., 2022) | 92.00 | 39.63 | 71.85 | 83.87 | 75.87 | -3.57 | 45.96 | 65.56 | 42.57 | 49.35 | -1.12 |
| FedProc (Mu et al., 2023) | 94.53 | 41.92 | 77.18 | 85.22 | 80.27 | 0.83 | 48.62 | 67.62 | 44.25 | 51.77 | 1.29 |
| FPL (Huang et al., 2023b) | 93.59 | 41.54 | 78.26 | 85.49 | 80.62 | 1.18 | 44.80 | 56.18 | 43.37 | 46.83 | -3.64 |
| FedCD (Long et al., 2023) | 93.11 | 40.11 | 72.71 | 84.88 | 76.79 | -2.65 | 48.66 | 69.41 | 45.07 | 52.25 | 1.77 |
| Fed3R (Fanì et al., 2024) | 93.49 | 40.29 | 73.65 | 85.75 | 78.75 | -0.69 | 47.12 | 53.79 | 42.23 | 47.82 | -2.66 |
| FedDIM (Liao et al., 2025) | **96.48** | 41.68 | 77.97 | 87.31 | 81.24 | 1.81 | 43.48 | 45.21 | 40.44 | 43.45 | -7.03 |
| FedDP (Fu et al., 2025) | 95.14 | 41.34 | 77.29 | 86.93 | 80.77 | 1.33 | 47.82 | 62.57 | 46.93 | 50.57 | 0.09 |
| FedGWC (Leo et al., 2025) | 93.24 | 40.76 | 76.23 | 86.21 | 79.62 | 0.18 | 46.44 | 67.45 | 46.31 | 50.49 | 0.01 |
| **FML** | 96.43 | **42.83** | **80.32** | **89.04** | **82.91** | **3.47** | **54.22** | **78.49** | **52.50** | **58.72** | **8.25** |

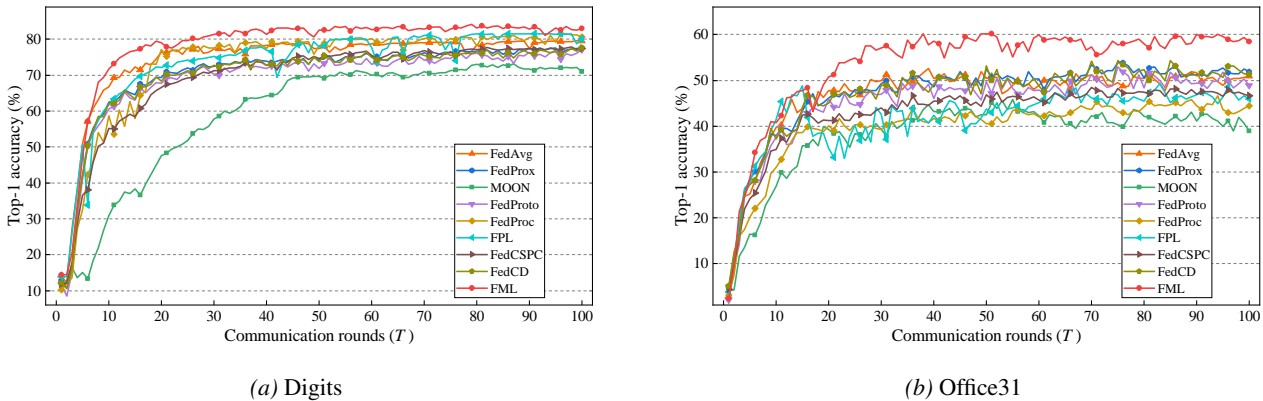

*(a)* Digits          *(b)* Office31

*Figure 2.* Comparison of average accuracy on different communication rounds with compared methods on Digits and Office31 tasks.

- Digits includes four domains: MNIST, USPS, SVHN and SYN, each covering digit categories from 0 to 9. MNIST and USPS datasets are comprised of grayscale images of handwritten digits. SVHN and SYN datasets contain color images, which are set against intricate and diverse backgrounds.

- Office31 includes three domains: Amazon, Webcam, and DSLR, each encompassing 31 categories. The primary differentiation among these domains lies in the method of image acquisition, the resulting quality of the images, and the diverse environmental conditions under which the images are captured, leading to varying levels of realism and clarity across the datasets.

To simulate a complex multi-domain federated learning scenario, we configure each client to possess data from two distinct domains. For the Digits task, we create six unique pairings from the four domains, assigning each to three clients, totaling 18 clients. For the Office31 task, three pairings are formed from the three domains, with

each assigned to three clients, making up nine clients in total. Considering task complexity and the scale of the data, clients randomly select varying amounts of local data from their domains. Unless otherwise specified, this paper defaults to setting the local data proportions for the Digits and Office31 tasks at 0.5% and 30%, respectively. We set a specific seed value to guarantee the replicability of results. The Digits task consists of four domains: MNIST, USPS, SVHN, and SYN, while the Office31 task includes three domains: Amazon, Webcam, and DSLR. For Digits and Office31 tasks, the experiments are conducted with ResNet-18 (He et al., 2016).

**Compared methods.** We compare our approach with a baseline FedAvg (AISTATS' 17 (McMahan et al., 2017)) and several SOTA federated learning methods that focus on the processing of heterogeneous data: FedProx (MLSys'20 (Li et al., 2020b)), MOON (CVPR'21 (Li et al., 2021)), Fed-Proto (AAAI'22 (Tan et al., 2022)), FedProc (FGCS'23 (Mu et al., 2023)), FPL (CVPR'23 (Huang et al., 2023b)), Fed-CSPC (MM'23 (Qi et al., 2023)), FedCD (MM'23 (Long

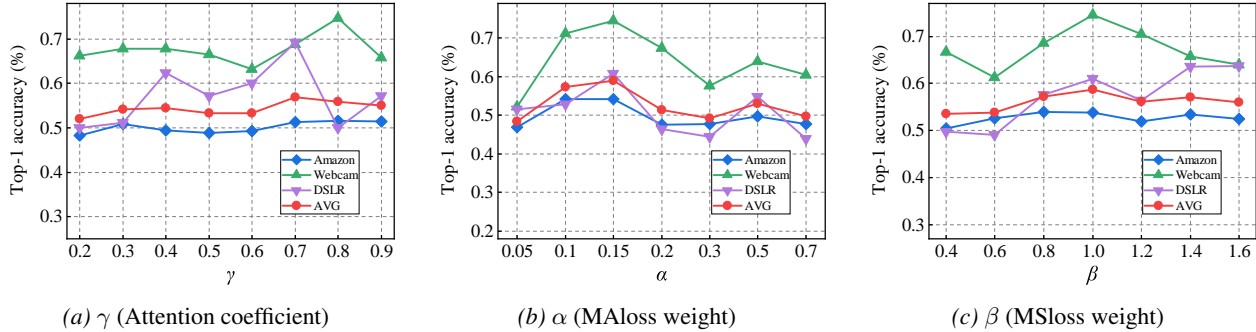

*(a)* $\gamma$ (Attention coefficient)      *(b)* $\alpha$ (MAloss weight)      *(c)* $\beta$ (MSloss weight)

*Figure 3.* The impact of hyperparameters $\gamma$, $\alpha$, and $\beta$ on Top-1 accuracy for the Office31 task. (a) The best AVG performance is obtained at $\gamma = 0.7$; (b) $\alpha$ yields the highest AVG accuracy at 0.15; (c) $\beta$ reaches the highest AVG accuracy at 1.0. See details in Sec. 4.3.2.

et al., 2023)), Fed3R (ICML'24(Fanì et al., 2024)), Fed-DIM (IJCAI'25(Liao et al., 2025)), FedDP (TMC '25(Fu et al., 2025)), FedGWC (ICML '25(Leo et al., 2025)). It is noteworthy that all methods employ the same network architecture to ensure a fair comparison.

**Implementation Details.** We implement FML by PyTorch (Paszke et al., 2019) and conduct experiments on machines running Ubuntu 18.04 and equipped with two NVIDIA GeForce RTX 3090 GPUs and an Intel Core(TM) i9-10900K CPU. For a fair comparison, we set the communication rounds $T = 100$ and the local epoch $E = 10$ by default. Under these settings, the accuracy of all methods can converge. We train the model using only CEloss for the initial 15 rounds. After the 15th round of training, the model acquires the feature extraction capability, and then the manifold guidance training method is introduced. We apply the Adam optimizer with a learning rate $lr = 0.01$ for all methods. The training batch size is set to 32. The feature vector dimension is 2048. Compared methods are implemented with their optimal parameters as reported in their respective publications. For the FML, the compression size is set to r=10 by default (analyzed in Sec. 4.3.1), the hyperparameters are set as follows: $\gamma = 0.7$, $\alpha = 0.15$, and $\beta = 1.0$ (analyzed in Sec. 4.3.2).

**Evaluation Metric.** Consistent with previous work (Mu et al., 2023; Huang et al., 2023b), we employ the Top-1 accuracy for fair evaluation. To evaluate performance across multiple domains, we use the "AVG" metric, a weighted average that accounts for the varying sizes of the test datasets in each domain. For the Digits task, weights are based on the relative sizes of the MNIST, SVHN, SYN, and USPS test sets, leading to respective contributions of 25%, 5%, 65%, and 5% to the AVG score. Similarly, for the Office31 task, the contributions from the Amazon, Webcam, and DSLR domains are calculated based on their test set sizes, resulting in proportions of approximately 68.4%, 19.4%, and 12.1% to the overall AVG metric. We conduct the experiments three times and consider the average accuracy of the last

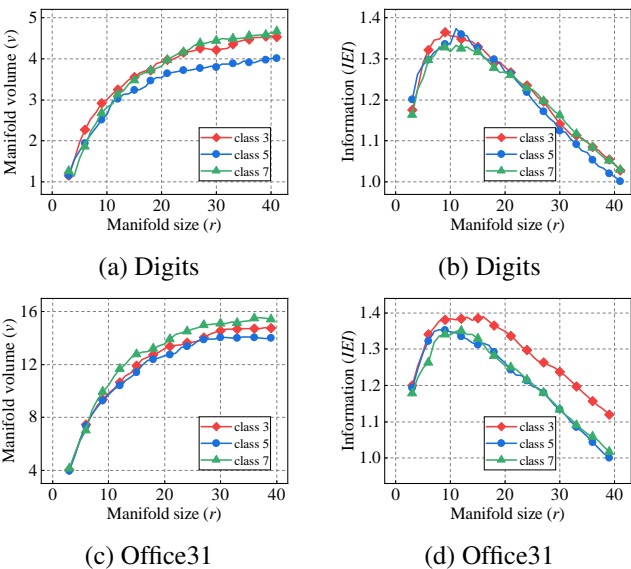

*(a)* Digits      *(b)* Digits

*(c)* Office31      *(d)* Office31

*Figure 4.* Analysis of the manifold size $r$ on the volume $v$ and information efficiency index (IEI) across classes 3, 5, and 7 for the Digits and Office31 tasks.

five communication rounds as the final result.

### 4.2. Comparison with State-of-the-Art Methods

Table 1 presents the Top-1 accuracy for all methods on two tasks, with FedAvg as the benchmark. The results highlight key insights: 1) FML significantly outperforms all methods, achieving an improvement of 1.67% on Digits and 6.47% on Office31 over the next best method; 2) FML achieves the highest accuracy in nearly all domains, demonstrating superior generalization and effectiveness across diverse domains; 3) Some methods perform worse than the baseline in specific configurations, due to their focus on label distribution heterogeneity in federated learning, which doesn't generalize well to domain heterogeneity. Figure 2 shows the average accuracy for all methods during each communication round. FML exhibits faster and more stable convergence, with a clear accuracy advantage in both tasks.

*Table 2.* Ablation studies of $\mathcal{L}_{MA}$ and $\mathcal{L}_{MS}$. Top: Results on Digits; Bottom: Results on Office31. The combination of both losses achieves the best performance.

| Components | | Digits Task | | | | |
|---|---|---|---|---|---|---|
| $\mathcal{L}_{MA}$ | $\mathcal{L}_{MS}$ | MNIST | SVHN | SYN | USPS | AVG |
| – | – | 93.67 | 40.28 | 76.57 | 84.76 | 79.44 |
| ✓ | – | 97.68 | 44.44 | 78.42 | 88.54 | 82.04 |
| – | ✓ | 96.48 | 41.68 | 77.97 | 87.31 | 81.24 |
| ✓ | ✓ | 96.43 | 42.83 | 80.32 | 89.04 | **82.91** |

| Components | | Office31 Task | | | |
|---|---|---|---|---|---|
| $\mathcal{L}_{MA}$ | $\mathcal{L}_{MS}$ | Amazon | Webcam | DSLR | AVG |
| – | – | 48.71 | 58.92 | 46.94 | 50.48 |
| ✓ | – | 52.80 | 62.48 | 53.40 | 54.75 |
| – | ✓ | 51.57 | 53.62 | 47.96 | 51.53 |
| ✓ | ✓ | 54.22 | 78.49 | 52.50 | **58.72** |

## 4.3. Evaluation of FML Details

### 4.3.1. SELECTION OF $r$ IN PMC

The parameter $r$ plays a significant role in influencing communication overhead and server computation speed. Our analysis investigates how varying values of $r$ affect the volume of the manifold, where an increase in volume represents an increase in information. We selected classes 3, 5, and 7 from the dataset for analysis. As shown in Figure 4(a) and (c), the volume of the manifold $v$ increases with rising values of $r$. However, as $r$ continues to escalate, the rate of volume $v$ growth begins to decelerate, indicating a reduction in the incremental gain of information. Furthermore, Figure 4(b) and (d) reveal the trend of the "Information Efficiency Index (IEI)" with the increase of $r$. The experiments show that near $r = 10$, the IEI reaches its maximum. This suggests that when $r$ is less than 10, the information quantity significantly increases with $r$, but beyond 10, the information gain provided by the additional samples is not significant. Therefore, selecting $r = 10$ achieves an optimal balance between manifold information quantity and computational overhead.

### 4.3.2. IMPACT OF HYPERPARAMETERS

FML incorporates three critical hyper-parameters: $\gamma$, $\alpha$ and $\beta$. We conduct the parameter sensitivity analysis on the Office31 task. $\gamma$ serves as a tuning parameter for the attention mechanism, varying within the range [0.2,0.9]. As shown in Figure 3a, FML achieves the best AVG when $\gamma$=0.7. $\alpha$ and $\beta$ are the weights of MAloss and MSloss respectively, used to adjust the impact of the loss function on overall training. Figure 3b reveals that the best AVG accuracy is achieved at $\alpha$ = 0.15, with the accuracy trends across individual domains aligning with the overall AVG trend. The hyperparameter $\beta$ directly affects the ability of $L_{MS}$ to achieve class separa-

tion. Figure 3c illustrates that an optimal outcome is attained when $\beta$ is set to 1.0. When $\beta$ exceeds 1, a slight fluctuation in AVG accuracy is observed, along with domain-specific variations; for instance, the Webcam domain experiences a decline in accuracy with an increase in $\beta$, while the DSLR domain exhibits an increase.

For fair cross-benchmark comparison, Table 1 reports results using a unified hyperparameter setting selected on Office31, while Appendix B provides dataset-specific sensitivity results.

### 4.3.3. LOSS FUNCTION ABLATIONS

We analyze the effectiveness of the proposed loss functions, $\mathcal{L}_{MA}$ and $\mathcal{L}_{MS}$, on Digits and Office31 datasets (Table 2). Compared to the FedAvg baseline, employing $\mathcal{L}_{MA}$ alone improves performance, confirming that guiding local training towards the global manifold enhances generalization. Similarly, using $\mathcal{L}_{MS}$ independently yields gains by improving class discrimination via manifold separation. Crucially, the combination of both losses in FML achieves the best results, validating our strategy of leveraging manifold transfer to boost model generalizability.

## 5. Conclusion

We propose Federated Manifold Learning (FML) to tackle domain heterogeneity in FL. By replacing prototypes with compressed manifold fusion via mutual learning, FML captures domain-invariant knowledge while preserving local characteristics. Experiments demonstrate significant performance gains over state-of-the-art methods. We believe this paradigm offers a promising direction for future work, including personalized FL and efficient interaction strategies.

## 6. Limitations and Future Work

The main focus of FML is structural knowledge transfer under domain heterogeneity rather than dedicated privacy protection. Although FML avoids sharing raw data, model updates and compressed manifold representations may still require additional safeguards in sensitive applications, such as differential privacy, secure aggregation, or multi-party computation. Moreover, FML currently adopts a uniform communication protocol across clients, which may be suboptimal in practical edge environments with heterogeneous bandwidth, computation, energy, and availability. Future work may develop resource-aware strategies that adapt upload frequency, manifold size, compression ratio, or client participation according to device constraints and the information contribution of local manifolds, thereby improving the scalability and deployability of FML in real-world federated edge learning systems.

## Acknowledgments

This work was supported in part by the National Key R&D Program of China (No. 2023YFB3107500), the National Natural Science Foundation of China (No. 62502362, No. 62502363, No. 62402358, No. 62220106004, No. 92467201), the China Postdoctoral Science Foundation (No. 2025M771506, No. 2025M781513), the Young Elite Scientist Sponsorship Program by CAST (No. YESS20240717), the Natural Science Basic Research Program of Shaanxi Province (No. 2025JC-YBQN-869), the Young Talent Fund of Association for Science and Technology in Shaanxi, China (No. 20240138), the "Pioneer" and "Leading Goose" R&D Program of Zhejiang (No. 2026SDXT014), the Open Topics from the Lion Rock Labs of Cyberspace Security (No. LRL24004), the Xidian University Specially Funded Project for Interdisciplinary Exploration (No. TZJHF202502), and the Fundamental Research Funds for the Central Universities (No. QTZX26022, No. QTZX26041, No. QTZX26050, No. QTZX26118).

## Impact Statement

This paper improves federated learning under domain heterogeneity by sharing compact structural knowledge across clients. FML may benefit applications where data come from different institutions, devices, or environments, such as healthcare, finance and mobile sensing.

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

# Federated Manifold Learning (FML): Tackling Domain Heterogeneity with Structural Knowledge Transfer

## Supplementary Material

## A. Detailed Allocation of Data

Here, we outline how data from each domain is allocated to individual clients, as shown in Figure 5.

## B. Additional Hyperparameter Sensitivity Analysis on Digits

To further evaluate the robustness of FML across different domain-heterogeneous benchmarks, we provide additional hyperparameter sensitivity analysis on the Digits benchmark. Specifically, we study the effects of the manifold mutual learning coefficient $\gamma$, the manifold approximation loss weight $\alpha$, and the manifold separation loss weight $\beta$. The results are reported in Tables 3–5. The best average performance on Digits is achieved with $\gamma = 0.9$, $\alpha = 0.7$, and $\beta = 0.8$. Compared with the optimal setting on Office31, i.e., $\gamma = 0.7$, $\alpha = 0.15$, and $\beta = 1.0$, we make the following observations. First, the optimal value of $\gamma$ remains relatively stable across benchmarks. The best performance is obtained with a large $\gamma$ on both Digits and Office31, indicating that cross-domain manifold fusion is a robust and effective component of FML. Second, the optimal value of $\alpha$ is higher on Digits. This may be because Digits contains larger visual domain gaps, such as the gap between grayscale handwritten digits in MNIST and complex colorful street-view digits in SVHN, thus requiring stronger manifold alignment. Third, the optimal values of $\beta$ are relatively close across the two benchmarks. The slightly larger value on Office31 may be attributed to its larger number of categories, i.e., 31 classes compared with 10 classes in Digits, which requires stronger inter-class separation. Overall, these results show that FML is not overly sensitive to a single hyperparameter setting. Although the optimal values vary across datasets, the performance remains consistently strong within a reasonable range, demonstrating the robustness of the proposed manifold-guided learning framework.

## C. Detailed Convergence Analysis of FML

In this section, we provide a detailed convergence analysis for the Federated Manifold Learning (FML) framework. Different from an idealized analysis that directly treats the server update as exact gradient descent, we explicitly model the effective update direction induced by manifold-guided local training, perceptual manifold compression, and attention-based manifold fusion.

*Table 3.* Sensitivity analysis of the manifold mutual learning coefficient $\gamma$ on the Digits benchmark. The reported results are average Top-1 accuracy (%).

| $\gamma$ | 0.2 | 0.3 | 0.4 | 0.5 | 0.6 | 0.7 | 0.8 | 0.9 | 1.0 |
|---|---|---|---|---|---|---|---|---|---|
| Acc. | 82.91 | 83.05 | 83.61 | 83.23 | 82.62 | 82.54 | 83.27 | **84.14** | 84.09 |

*Table 4.* Sensitivity analysis of the manifold approximation loss weight $\alpha$ on the Digits benchmark.

| $\alpha$ | 0.05 | 0.1 | 0.15 | 0.2 | 0.3 | 0.5 | 0.7 | 0.9 |
|---|---|---|---|---|---|---|---|---|
| Acc. | 84.06 | 82.59 | 82.54 | 83.32 | 82.41 | 83.46 | **84.25** | 83.90 |

*Table 5.* Sensitivity analysis of the manifold separation loss weight $\beta$ on the Digits benchmark.

| $\beta$ | 0.4 | 0.6 | 0.8 | 1.0 | 1.2 | 1.4 |
|---|---|---|---|---|---|---|
| Acc. | 83.62 | 82.86 | **84.01** | 82.54 | 82.27 | 83.64 |

The global objective of the federated system is to minimize

$$\min_w \mathcal{F}(w) \triangleq \sum_{m=1}^{M} p_m \mathcal{F}_m(w), \tag{6}$$

where $w \in \mathbb{R}^d$ denotes the model parameters, $p_m = \frac{|\mathcal{D}_m|}{\sum_{k=1}^{M} |\mathcal{D}_k|}$ is the weight of client $m$, and $\mathcal{F}_m(w)$ is the expected local loss over the data distribution of client $m$.

In FML, the local objective is a composite loss:

$$\mathcal{F}_m(w) = \mathbb{E}_{(x,y)\sim\mathcal{D}_m}\Big[\mathcal{L}_{\text{CE}}(w; x, y) + \alpha\mathcal{L}_{\text{MA}}(w; x, y)$$
$$+ \beta\mathcal{L}_{\text{MS}}(w; x, y)\Big]. \tag{7}$$

Here, $\mathcal{L}_{\text{CE}}$ is the standard cross-entropy loss, $\mathcal{L}_{\text{MA}}$ is the manifold approximation loss, and $\mathcal{L}_{\text{MS}}$ is the manifold separation loss.

### C.1. Assumptions

Our analysis relies on the following standard assumptions.

**Assumption C.1** ($L$-Smoothness)**.** The global objective $\mathcal{F}(w)$ is continuously differentiable and $L$-smooth. That is, for any $w_1, w_2 \in \mathbb{R}^d$,

$$\mathcal{F}(w_2) \le \mathcal{F}(w_1) + \langle\nabla\mathcal{F}(w_1), w_2 - w_1\rangle$$
$$+ \frac{L}{2}\|w_2 - w_1\|^2. \tag{8}$$

**Assumption C.2** (Local Polyak–Lojasiewicz Condition)**.** Along the optimization trajectory considered in this anal-

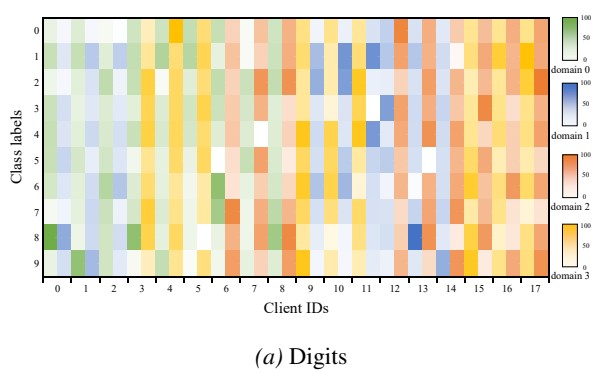

*(a)* Digits

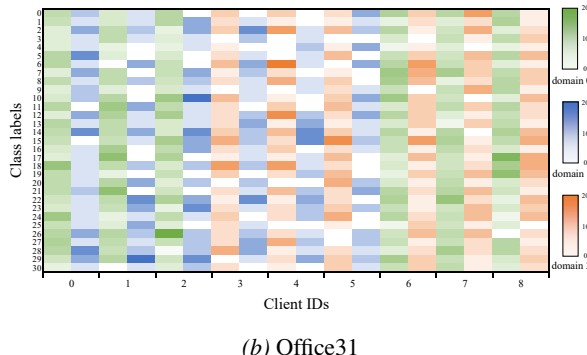

*(b)* Office31

*Figure 5.* Distribution of client data in the Digits and Office31 tasks. For the Digits task, MNIST, USPS, SVHN, and SYN represent the four domains, while for the Office31 task, Amazon, Webcam, and DSLR constitute the three domains.

ysis, the global objective $\mathcal{F}(w)$ satisfies a local Polyak–Lojasiewicz (PL) condition with constant $\mu > 0$:

$$\frac{1}{2}\left\|\nabla\mathcal{F}(w)\right\|^2 \geq \mu\left(\mathcal{F}(w) - \mathcal{F}^\star\right), \qquad (9)$$

where $\mathcal{F}^\star$ denotes the optimal objective value.

**Applicability to FML's Objective.**  The FML objective contains the cross-entropy loss, the manifold approximation loss, and the manifold separation loss. Since the feature extractor is parameterized by a deep neural network, the resulting objective is generally non-convex. Therefore, instead of requiring global strong convexity, we adopt a local PL-type condition along the optimization trajectory. This condition is commonly used to establish linear convergence for non-convex objectives in a local well-behaved region. Under this condition, our analysis focuses on how the proposed compression and fusion mechanisms affect the effective update direction and the convergence rate.

### C.2. Mechanism-Aware Convergence Theorem

We now prove a mechanism-aware convergence result for the global model sequence generated by FML.

**Assumption C.3** (Mechanism-Induced Direction Deviation). At communication round $t$, the server-side model update of FML can be written in the effective form

$$w^{t+1} = w^t - \eta d_t, \qquad (10)$$

where $d_t$ denotes the effective model-update direction induced by manifold-guided local training and server-side aggregation. The compressed and fused manifolds affect $d_t$ through the local objective in subsequent client updates.

We assume that the deviation between $d_t$ and the true global gradient is bounded by

$$\left\|d_t - \nabla\mathcal{F}(w^t)\right\| \leq \delta\left\|\nabla\mathcal{F}(w^t)\right\|, \quad \delta = \delta_{\text{comp}} + \delta_{\text{att}} < 1. \qquad (11)$$

Here, $\delta_{\text{comp}}$ characterizes the distortion introduced by perceptual manifold compression, and $\delta_{\text{att}}$ characterizes the bias introduced by attention-based manifold fusion.

Define

$$A_\delta = (1 - \delta) - \frac{L\eta}{2}(1 + \delta)^2. \qquad (12)$$

**Theorem C.4** (Mechanism-Aware Linear Convergence of FML). *Suppose Assumptions C.1, C.2, and C.3 hold. If*

$$0 < \eta < \frac{2(1 - \delta)}{L(1 + \delta)^2} \qquad (13)$$

*and* $2\mu\eta A_\delta < 1$, *then for any* $T \geq 1$,

$$\mathcal{F}(w^T) - \mathcal{F}^\star \leq \rho^T\left[\mathcal{F}(w^0) - \mathcal{F}^\star\right], \qquad (14)$$

*where*

$$\rho = 1 - 2\mu\eta A_\delta \in (0, 1). \qquad (15)$$

*Proof.* Let

$$g_t = \nabla\mathcal{F}(w^t). \qquad (16)$$

By Eq. (10),

$$w^{t+1} - w^t = -\eta d_t. \qquad (17)$$

By $L$-smoothness, we have

$$\mathcal{F}(w^{t+1}) \leq \mathcal{F}(w^t) + \left\langle g_t, w^{t+1} - w^t\right\rangle + \frac{L}{2}\left\|w^{t+1} - w^t\right\|^2$$

$$= \mathcal{F}(w^t) - \eta\left\langle g_t, d_t\right\rangle + \frac{L\eta^2}{2}\left\|d_t\right\|^2. \qquad (18)$$

Define the direction error as

$$e_t = d_t - g_t. \qquad (19)$$

By Assumption C.3,

$$\left\|e_t\right\| = \left\|d_t - g_t\right\| \leq \delta\|g_t\|. \qquad (20)$$

We first lower-bound the descent alignment term:

$$
\begin{aligned}
\langle g_t, d_t \rangle &= \langle g_t, g_t + e_t \rangle \\
&= \|g_t\|^2 + \langle g_t, e_t \rangle \\
&\geq \|g_t\|^2 - \|g_t\|\|e_t\| \\
&\geq (1 - \delta)\|g_t\|^2.
\end{aligned}
\tag{21}
$$

We also upper-bound the norm of the effective update direction:

$$
\begin{aligned}
\|d_t\| &= \|g_t + e_t\| \\
&\leq \|g_t\| + \|e_t\| \\
&\leq (1 + \delta)\|g_t\|.
\end{aligned}
\tag{22}
$$

Thus,
$$
\|d_t\|^2 \leq (1 + \delta)^2 \|g_t\|^2.
\tag{23}
$$

Substituting Eq. (21) and Eq. (23) into Eq. (18), we obtain

$$
\mathcal{F}(w^{t+1}) \leq \mathcal{F}(w^t) - \eta A_\delta \|g_t\|^2.
\tag{24}
$$

The step-size condition in Eq. (13) implies $A_\delta > 0$, so the effective update is a descent step. By the local PL condition in Assumption C.2,

$$
\|g_t\|^2 = \left\|\nabla \mathcal{F}(w^t)\right\|^2 \geq 2\mu \left(\mathcal{F}(w^t) - \mathcal{F}^\star\right).
\tag{25}
$$

Combining Eq. (24) and Eq. (25), we have

$$
\mathcal{F}(w^{t+1}) \leq \mathcal{F}(w^t) - 2\mu\eta A_\delta \left(\mathcal{F}(w^t) - \mathcal{F}^\star\right).
\tag{26}
$$

Let
$$
\theta^t = \mathcal{F}(w^t) - \mathcal{F}^\star
\tag{27}
$$

denote the optimality gap. Then Eq. (26) implies

$$
\theta^{t+1} \leq (1 - 2\mu\eta A_\delta)\, \theta^t.
\tag{28}
$$

Define
$$
\rho = 1 - 2\mu\eta A_\delta.
\tag{29}
$$

Since $A_\delta > 0$ and $2\mu\eta A_\delta < 1$, we have $\rho \in (0, 1)$. Therefore,
$$
\theta^{t+1} \leq \rho \theta^t.
\tag{30}
$$

Recursively applying this inequality from $t = 0$ to $T - 1$ gives
$$
\theta^T \leq \rho^T \theta^0.
\tag{31}
$$

Substituting back the definition of $\theta^t$, we obtain

$$
\mathcal{F}(w^T) - \mathcal{F}^\star \leq \rho^T \left[\mathcal{F}(w^0) - \mathcal{F}^\star\right].
\tag{32}
$$

This completes the proof. $\square$

**Connection to the idealized case.** When $\delta_{\text{comp}} = \delta_{\text{att}} = 0$, we have $\delta = 0$, $A_\delta = 1 - \frac{L\eta}{2}$, and

$$
\rho = 1 - 2\mu\eta \left(1 - \frac{L\eta}{2}\right),
\tag{33}
$$

which recovers the idealized gradient-descent convergence rate. Thus, the idealized convergence result can be viewed as a special case of the mechanism-aware theorem.

