# Federated Manifold Learning (FML): Tackling Domain Heterogeneity with Structural Knowledge Transfer

## Supplementary Material

## A. Detailed Allocation of Data

Here, we outline how data from each domain is allocated to individual clients, as shown in Figure 5.

## B. Detailed Convergence Analysis of FML

In this section, we provide a detailed convergence analysis for the Federated Manifold Learning (FML) framework presented in the main paper. Our objective is to demonstrate that, under standard assumptions in convex optimization, the global model generated by FML converges linearly to the optimal solution.

The global objective of the federated system is to minimize the global loss function $\mathcal{F}(w)$, which is a weighted average of the local loss functions of the $M$ clients:

$$\min_w \mathcal{F}(w) \triangleq \sum_{m=1}^{M} p_m \mathcal{F}_m(w), \qquad (7)$$

where $w \in \mathbb{R}^d$ represents the model parameters, $p_m = \frac{|\mathcal{D}_m|}{\sum_{k=1}^{M} |\mathcal{D}_k|}$ is the weight for client $m$, and $\mathcal{F}_m(w)$ is the expected local loss for client $m$ over its data distribution.

In our FML framework, the local loss function for client $m$ is a composite loss:

$$\mathcal{F}_m(w) = \mathbb{E}_{(x,y)\sim\mathcal{D}_m}\Big[\mathcal{L}_{CE}(w;x,y) + \alpha\mathcal{L}_{MA}(w;x,y)$$
$$+ \beta\mathcal{L}_{MS}(w;x,y)\Big]. \quad (8)$$

For the purpose of this theoretical analysis, we consider a simplified, idealized version of the FML update process that mirrors centralized gradient descent, where the update is based on the true global gradient. This is a common practice for analyzing the foundational stability of FL algorithms. The global model at communication round $t + 1$ is updated as:

$$w^{t+1} = w^t - \eta \nabla \mathcal{F}(w^t), \qquad (9)$$

where $\eta$ is the learning rate. We will now show that this process converges linearly.

### B.1. Assumptions

Our proof relies on the following standard assumptions, which are common in the analysis of large-scale optimization algorithms.

**Assumption B.1** ($L$-Smoothness)**.** The global loss function $\mathcal{F}(w)$ is continuously differentiable and $L$-smooth for some constant $L > 0$. This means that its gradient $\nabla\mathcal{F}$ is Lipschitz continuous with constant $L$. For any $w_1, w_2 \in \mathbb{R}^d$, this implies:

$$\mathcal{F}(w_2) \leq \mathcal{F}(w_1) + \langle \nabla\mathcal{F}(w_1), w_2 - w_1 \rangle + \frac{L}{2}\|w_2 - w_1\|^2. \tag{10}$$

Intuitively, this assumption means the curvature of the loss function is bounded, preventing the gradient from changing arbitrarily fast.

**Assumption B.2** ($\mu$-Strong Convexity)**.** The global loss function $\mathcal{F}(w)$ is $\mu$-strongly convex for some constant $\mu > 0$. For any $w_1, w_2 \in \mathbb{R}^d$, this is defined as:

$$\mathcal{F}(w_2) \geq \mathcal{F}(w_1) + \langle \nabla\mathcal{F}(w_1), w_2 - w_1 \rangle + \frac{\mu}{2}\|w_2 - w_1\|^2. \tag{11}$$

This assumption guarantees that the function has a unique minimizer $w^* = \arg\min_w \mathcal{F}(w)$ and that the function's shape is sufficiently "bowl-shaped" everywhere.

**Applicability to FML's Loss Function.** We briefly justify why these assumptions are reasonable for our composite loss. The Cross-Entropy loss ($\mathcal{L}_{CE}$) is convex (though not strongly convex) and smooth. The Manifold Approximation Loss ($\mathcal{L}_{MA}$), being a sum of squared $\ell_2$-norms, is both smooth and convex. The Manifold Separation Loss ($\mathcal{L}_{MS}$), a form of contrastive loss, is generally non-convex. However, in deep learning theory, it is common to assume that the overall objective function is $\mu$-strongly convex within a relevant neighborhood of the optimal solution, or that a regularization term is added to ensure this property. For the purpose of this analysis, we assume the composite function $\mathcal{F}(w)$ satisfies these standard conditions, allowing us to establish a foundational convergence guarantee.

### B.2. Main Convergence Theorem

We now restate and prove the main theorem regarding the linear convergence of the FML global model sequence.

**Theorem B.3** (Linear Convergence of FML)**.** *Suppose Assumptions B.1 and B.2 hold. Let $\{w^t\}$ be the sequence of global models generated by the idealized FML update with a fixed learning rate $\eta$ satisfying $0 < \eta \leq \frac{1}{L}$. Then, the optimality gap at round $T$ is bounded by:*

$$\mathcal{F}(w^T) - \mathcal{F}(w^*) \leq \rho^T \left[\mathcal{F}(w^0) - \mathcal{F}(w^*)\right], \qquad (12)$$

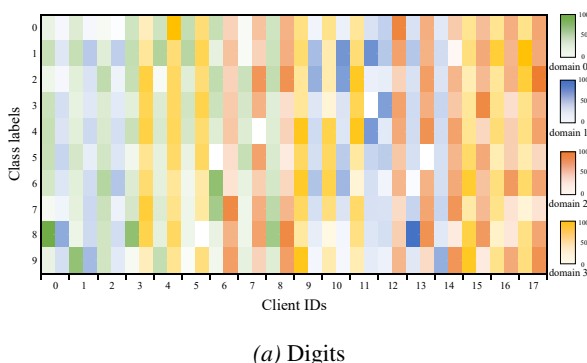

*(a)* Digits

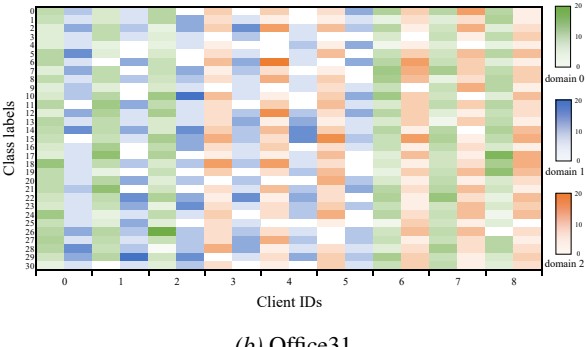

*(b)* Office31

*Figure 5.* Distribution of client data in the Digits and Office31 tasks. For the Digits task, MNIST, USPS, SVHN, and SYN represent the four domains, while for the Office31 task, Amazon, Webcam, and DSLR constitute the three domains.

*where $\rho = (1 - \mu\eta)$ is the convergence rate, demonstrating a linear (or geometric) decay of the error. A tighter bound for $\rho$ is also derived in the proof.*

*Proof.* Let $w^*$ be the unique minimizer of $\mathcal{F}(w)$, where $\nabla\mathcal{F}(w^*) = 0$.

Our analysis begins with the $L$-smoothness property from Assumption B.1. Let's apply it to the model at round $t + 1$, denoted as $w^{t+1}$:

$$\mathcal{F}(w^{t+1}) \leq \mathcal{F}(w^t) + \langle \nabla\mathcal{F}(w^t), w^{t+1} - w^t \rangle \\ + \frac{L}{2}\|w^{t+1} - w^t\|^2 \quad (13)$$

Now, we substitute the gradient descent update rule, $w^{t+1} - w^t = -\eta\nabla\mathcal{F}(w^t)$, into the inequality (13):

$$\mathcal{F}(w^{t+1}) \leq \mathcal{F}(w^t) + \langle \nabla\mathcal{F}(w^t), -\eta\nabla\mathcal{F}(w^t) \rangle \\ + \frac{L}{2}\| -\eta\nabla\mathcal{F}(w^t)\|^2 \quad (14)$$

$$= \mathcal{F}(w^t) - \eta\langle \nabla\mathcal{F}(w^t), \nabla\mathcal{F}(w^t) \rangle \\ + \frac{L\eta^2}{2}\|\nabla\mathcal{F}(w^t)\|^2 \quad (15)$$

$$= \mathcal{F}(w^t) - \eta\|\nabla\mathcal{F}(w^t)\|^2 + \frac{L\eta^2}{2}\|\nabla\mathcal{F}(w^t)\|^2 \quad (16)$$

$$= \mathcal{F}(w^t) - \eta\left(1 - \frac{L\eta}{2}\right)\|\nabla\mathcal{F}(w^t)\|^2 \quad (17)$$

Next, we leverage the $\mu$-strong convexity from Assumption B.2. A standard result derived from strong convexity is that for any $w$:

$$\|\nabla\mathcal{F}(w)\|^2 \geq 2\mu(\mathcal{F}(w) - \mathcal{F}(w^*)). \quad (18)$$

We can now substitute this bound for the squared gradient norm (18) into our progress inequality (17):

$$\mathcal{F}(w^{t+1}) \leq \mathcal{F}(w^t) - \eta\left(1 - \frac{L\eta}{2}\right) \\ \times \left[2\mu(\mathcal{F}(w^t) - \mathcal{F}(w^*))\right] \quad (19)$$

$$= \mathcal{F}(w^t) - 2\mu\eta\left(1 - \frac{L\eta}{2}\right) \\ \times (\mathcal{F}(w^t) - \mathcal{F}(w^*)) \quad (20)$$

To simplify the expression, let's subtract the optimal value $\mathcal{F}(w^*)$ from both sides of the inequality:

$$\mathcal{F}(w^{t+1}) - \mathcal{F}(w^*) \leq \mathcal{F}(w^t) - \mathcal{F}(w^*) \\ - 2\mu\eta\left(1 - \frac{L\eta}{2}\right)(\mathcal{F}(w^t) - \mathcal{F}(w^*)) \quad (21)$$

$$= \left[1 - 2\mu\eta\left(1 - \frac{L\eta}{2}\right)\right] \\ \times (\mathcal{F}(w^t) - \mathcal{F}(w^*)) \quad (22)$$

Let $\theta^t = \mathcal{F}(w^t) - \mathcal{F}(w^*)$ denote the optimality gap at round $t$. The recursive relationship from (22) can be written as:

$$\theta^{t+1} \leq \rho \cdot \theta^t, \quad (23)$$

where the convergence rate $\rho = 1 - 2\mu\eta\left(1 - \frac{L\eta}{2}\right)$.

For convergence, we need $\rho < 1$. Since $\mu, \eta > 0$ and our choice of learning rate $\eta \leq \frac{1}{L}$ ensures that $(1 - \frac{L\eta}{2}) \geq \frac{1}{2}$, the term $2\mu\eta(1 - \frac{L\eta}{2})$ is strictly positive, thus $\rho < 1$. If we choose a simpler (though slightly looser) step size $\eta = 1/L$, the rate becomes $\rho = 1 - \mu/L$. A common step size is $\eta \approx 1/L$, which leads to $\rho \approx 1 - \mu/L < 1$. Let's use the tighter bound derived above.

By applying the recursive relationship (22) for $T$ rounds,

starting from $t = 0$:

$$\theta^T \leq \rho \cdot \theta^{T-1} \tag{24}$$

$$\leq \rho^2 \cdot \theta^{T-2} \tag{25}$$

$$\vdots \tag{26}$$

$$\leq \rho^T \cdot \theta^0 \tag{27}$$

Substituting back the definition of $\theta^t$, we arrive at the final result:

$$\mathcal{F}(w^T) - \mathcal{F}(w^*) \leq \left[1 - 2\mu\eta\left(1 - \frac{L\eta}{2}\right)\right]^T \tag{28}$$
$$\times \left[\mathcal{F}(w^0) - \mathcal{F}(w^*)\right].$$

$\square$

This completes the proof. We have shown that the sub-optimality gap decreases geometrically at each round, which is the definition of linear convergence. This result provides theoretical grounding for the stability and effectiveness of the FML optimization process.