# OpenReview forum: "Federated Manifold Learning (FML): Tackling Domain Heterogeneity with Structural Knowledge Transfer"
_ICML.cc/2026/Conference — ICML 2026 regular_

### Official Review · Reviewer_yUKK · 2026-03-09

**Soundness:** 3
**Presentation:** 3
**Significance:** 3
**Originality:** 2
**Overall Recommendation:** 3
**Confidence:** 4

**Summary:**

This paper studies federated learning under domain heterogeneity, where clients possess data from different visual domains and standard parameter averaging leads to poor global generalization. The authors argue that existing prototype-based approaches are limited because representing each class with a single prototype fails to capture the structural variability of feature distributions across domains. To address this limitation, the paper proposes Federated Manifold Learning (FML). Instead of exchanging prototypes, clients transmit compressed manifold representations extracted from their local feature spaces using a DPP-based sampling strategy. On the server, a Manifold Mutual Learning (MML) module performs clustering and attention-based fusion to integrate cross-domain structural information. The resulting global manifolds are then used to guide local training via manifold approximation and separation losses. Experiments on Digits and Office31 show improvements over several federated learning baselines, and the paper includes ablation studies analyzing the impact of the manifold compression and loss components.

**Compliance With Llm Reviewing Policy:**

Affirmed.

**Key Questions For Authors:**

•	The experiments are conducted only on Digits and Office31. Could the authors evaluate the method on larger and more diverse datasets to demonstrate robustness and scalability.
•	Since the proposed method exchanges compressed manifolds between clients and the server, how does the communication cost compare with standard federated learning methods such as FedAvg or prototype-based approaches.
•	The theoretical analysis assumes standard convexity conditions like FedAvg. How does the manifold compression and attention-based fusion affect convergence in practice.
•	The authors provide additional insight into how the manifold representations behave across highly heterogeneous domains.

**Limitations:**

The paper partially discusses limitations but could elaborate more explicitly. In particular:
•	This approach may introduce additional communication overhead due to the transmission of manifold representations.
•	The method relies on feature representations from deep models, which may behave differently across datasets.
•	Experiments are conducted on relatively small datasets, so the scalability of the method remains unclear.

**Strengths And Weaknesses:**

Strengths
•	Domain heterogeneity remains one of the main challenges in federated learning. Since clients often collect data under different conditions, addressing domain shifts is important for making federate learning systems work reliably in practice.
•	The idea of representing class information as manifold structures rather than single prototypes is intuitive. This perspective makes sense because feature distributions across domains can be complex, and manifolds may capture this structure better than a single representative vector.
•	The proposed pipeline (PMC, MML, and manifold-guided training) is clearly defined and the algorithmic workflow is easy to follow.
•	The experimental section includes comparisons with several federated learning baselines and provides ablation studies. These analyses help illustrate how the proposed losses and manifold compression mechanism contribute to the final performance.





Weaknesses
•	While the idea is interesting, the overall methodological contribution appears somewhat incremental. The framework mainly combines existing concepts such as manifold representations, clustering, attention-based aggregation, and contrastive-style losses within a federated learning setting.
•	Evaluation is restricted to Digits and Office31, which are relatively small benchmarks. Experiments on larger and more diverse datasets would strengthen the claims.
•	Since the method exchanges manifold representations, the paper should discuss or measure communication overhead and scalability relative to existing FL methods.
•	The convergence discussion relies on standard assumptions like FedAvg and does not provide deeper insight into the proposed manifold-based updates.

---

> ### Author Rebuttal · Authors · 2026-03-31
>
> Dear Reviewer yUKK,
>
> Thank you for your thoughtful review and valuable feedback. We appreciate your positive comments on the motivation and empirical analysis of our method, as well as your constructive suggestions on its limitations. We have carefully considered all of your concerns and address them point by point below.
>
> ## Regarding Weakness 1: Methodological Novelty.
>
> Thank you for recognizing the interest of the core idea and for raising this important concern about methodological novelty. We agree that some components in FML are not new individually. However, the contribution of FML lies in a new federated learning paradigm: shifting from point-wise prototype sharing to structure-level manifold transfer. By compressing, fusing, and feeding back class manifolds, FML establishes a new structural communication and learning framework for heterogeneous federated learning, rather than merely combining existing techniques.
>
> ## Regarding Weakness 2 & Questions 1 & Limitations 2,3: Evaluation on Larger and More Diverse Benchmarks.
>
> Your suggestion on evaluating larger models and datasets is highly valuable. In the original manuscript, we used Digits, Office31, and ResNet to follow the standard evaluation protocols in prior heterogeneous federated learning studies, ensuring fair comparison. These settings are consistent with mainstream practice in the literature:
>
> [1] Geometric knowledge-guided localized global distribution alignment for federated learning. CVPR, 2025.
>
> [2] Re-fed+: A better replay strategy for federated incremental learning. IEEE TPAMI, 2025.
>
> Meanwhile, we additionally conduct experiments on the  larger-scale OfficeHome benchmark and with the stronger ResNet-50 backbone. The results show that our method remains effective under more challenging settings.
>
> | Methods | Art | Clipart | Product | Real World | AVG |
> |---|---:|---:|---:|---:|---:|
> | FedAvg | 55.18 | 46.27 | 73.51 | 66.28 | 60.31 |
> | FedProto | 54.42 | 46.92 | 72.61 | 64.85 | 59.70 |
> | FML | 62.12 | 58.39 | 77.29 | 76.27 | 68.52 |
>
> ## Regarding Weakness 3 & Questions 2 & Limitations 1: Communication Cost.
>
> This is an excellent practical question. We're glad you asked, as this optimization step is indeed a core component of our calibration mechanism. A similar concern about its computational impact was raised by another reviewer, which prompted us to perform a detailed analysis. We are happy to share the results, which confirm that this process is efficient in practice.
>
> We conducted a quantitative analysis to measure th e communication overhead of our method. The table below compares the time cost per communication round of FedAvg, FedProto, and FML. The results show that FML introduces additional communication overhead compared with FedAvg, as expected, but remains comparable to prototype-based methods while providing stronger performance gains.
>
> | Methods | Digits | Office31 | OfficeHome |
> |---|---:|---:|---:|
> | FedAvg | 7.54s | 7.54s | 7.55s |
> | FedProto | 7.56s | 7.56s | 7.57s |
> | FML | 7.62s | 7.72s | 7.89s |
>
> ## Regarding Weakness 4 & Questions 3: Convergence Discussion.
>
> Thank you for this insightful comment. It is highly valuable for improving the theoretical clarity and completeness of the manuscript. In the revision, we will add a mechanism-aware convergence result for the server-side FML update $w_{t+1}=w_t-\eta d_t,$ where $d_t$  is the compressed-and-fused aggregation direction constructed from client updates computed on the full local objective. Specifically, under L-smoothness and a local Polyak–Łojasiewicz condition, if the deviation of  $d_t$  from the true global gradient is bounded as $\lVert d_t-\nabla F(w_t)\rVert \le (\delta_{\mathrm{comp}}+\delta_{\mathrm{att}})\lVert \nabla F(w_t)\rVert = \delta \lVert \nabla F(w_t)\rVert,\quad \delta<1,$ then for $0<\eta<\frac{2(1-\delta)}{L(1+\delta)^2},$ we obtain $F(w_{t+1})-F^\star \le \rho(F(w_t)-F^\star),$ with $\rho = 1-2\mu\eta\left[(1-\delta)-\frac{L\eta}{2}(1+\delta)^2\right] < 1.$
>
> This result clarifies the roles of the two modules: manifold compression affects convergence through $\delta_{\mathrm{comp}}$, while attention-based fusion affects it through $\delta_{\mathrm{att}}$. Larger distortion or bias leads to a larger $\delta$ and slower convergence, with the base theorem recovered as the special case $\delta=0$.
>
> ## Regarding Questions 4: Manifold Behavior Analysis.
>
> Thank you for this insightful question. To clarify manifold behavior under domain heterogeneity, we provide additional qualitative analysis on the Digits tasks by comparing the feature distributions of FedAvg and FML in Figure 1–4 (https://anonymous.4open.science/r/rebuttal-Anonymization/Anonymous_figure.pdf). FML produces more structured feature distributions: samples from different domains are better aligned within the same class, and samples from different classes remain clearly separated. This further supports the effectiveness of FML in capturing transferable structure-level knowledge.

---

> > ### Author Rebuttal · Reviewer_yUKK · 2026-04-05
> >
> > The rebuttal does not adequately address the other issues I raised. For these reasons, I will maintain my current score.

---

> > > ### Author Response · Authors · 2026-04-06
> > >
> > > ## Response to Reviewer yUKK
> > > Thank you very much for taking the time to read our rebuttal and for clarifying that some concerns remain unresolved from your perspective. We sincerely appreciate your careful evaluation of our submission.
> > >
> > > We understand that, due to the short rebuttal format, some of our responses may have been too concise. If our previous explanation was not sufficiently clear, we would like to briefly restate the key points here. Our intention was not to avoid the remaining issues, but rather to address them within the limited space available.
> > >
> > > In particular, we would like to emphasize the following:
> > > 1. Methodological novelty：We agree that some individual components of FML are related to existing techniques. Our main contribution, however, is not a standalone module, but a new heterogeneous federated learning paradigm: shifting from point-wise prototype sharing to structure-level manifold transfer. Through manifold compression, fusion, and feedback, FML enables structural communication across clients rather than a simple combination of prior components.
> > > 2. Evaluation on larger settings：In addition to the standard heterogeneous FL benchmarks used in the submission, we further conducted experiments on the larger-scale OfficeHome benchmark with a stronger ResNet-50 backbone. The additional results show that FML remains effective under more challenging settings, with clear gains over representative baselines.
> > > 3. Communication cost：We agree that this is an important practical concern. Our additional quantitative analysis shows that FML introduces some overhead compared with FedAvg, which is expected, but its per-round communication cost remains comparable to prototype-based methods such as FedProto, while achieving substantially stronger performance.
> > > 4. Convergence insight specific to the proposed update：We agree that a standard FedAvg-style result alone does not explain the manifold-based update. In the rebuttal, we therefore added a mechanism-aware discussion for the FML update $w_{t+1} = w_t - \eta d_t$, where the deviation of $d_t$ from the true global gradient is decomposed into two method-specific terms: compression distortion $\delta_{\mathrm{comp}}$ and fusion bias $\delta_{\mathrm{att}}$. This shows that larger distortion or bias leads to slower convergence, while the FedAvg-style result is recovered when $\delta = 0$.
> > >
> > > 5. Manifold behavior analysis：To further clarify the mechanism of FML under domain heterogeneity, we additionally provided qualitative feature-distribution analysis on Digits by comparing FedAvg and FML. The visualizations show that FML yields more structured representations: samples from different domains are better aligned within the same class, while samples from different classes remain more clearly separated. We believe this provides direct evidence that FML captures transferable structure-level knowledge, which is precisely the central motivation of the method.
> > >
> > > We fully respect your decision to maintain your score, and we are grateful for your thoughtful comments, which helped us improve the presentation and clarify the scope of our claims. We hope the AC will consider both the remaining concerns and the above clarifications when making the final assessment. If accepted, we will incorporate all of these points carefully into the final version to further strengthen the paper.
> > >
> > > Thank you again for your time and consideration.

---

### Official Review · Reviewer_WcQ8 · 2026-03-11

**Soundness:** 3
**Presentation:** 3
**Significance:** 3
**Originality:** 3
**Overall Recommendation:** 4
**Confidence:** 3

**Summary:**

FML is proposed to address domain heterogeneity between clients using perceptual manifolds contributed by each client to guide training. Individual client manifolds are aggregated into a global manifold using FINCH clustering and attention between clusters. The local training objective is modified to consider the global manifold via the addition of two additional terms, manifold separation and approximation loss, in addition to CE loss. The additional communication burden of communicating the manifolds is addressed by manifold compression.

**Compliance With Llm Reviewing Policy:**

Affirmed.

**Key Questions For Authors:**

1. The global manifold corrections to the local might significantly help to combat client drift. How does performance hold up against increasing numbers of local gradient steps?

**Limitations:**

yes

**Strengths And Weaknesses:**

Strengths:

* The paper consists of many different concepts that are well organized and explained, making the ideas easy to follow
* There are many baseline comparisons
* Ablation studying how compression affects performance
* Manifolds are a good strategy of obtaining richer information about client data distributions. The process for combining the manifolds is well justified

Weaknesses:

* There is no discussion on related work using manifolds to address client data heterogeneity in FL
* The results on the digits data are significantly weaker than for office31. Are the results for only one seed? An average with some sort of variance measurement might provide a stronger indicator of performance.
* 18 clients is relatively small scale for FL, an ablation where the number of clients is increased would strengthen this work

---

> ### Author Rebuttal · Authors · 2026-03-31
>
> Dear Reviewer WcQ8,
>
> We sincerely thank you for your thoughtful and constructive review. We appreciate your positive comments on the clarity of the paper, the baseline comparisons, the ablation study, and the motivation of using manifolds to model client heterogeneity.
>
> Your comments are very helpful for improving the paper. We address them in detail below, and the corresponding clarifications and revisions will be incorporated into the final manuscript.
>
> ## Regarding Weakness 1: Related Work.
> Thank you for pointing this out. We agree that this discussion is helpful. Accordingly, we add a brief discussion of manifold-based federated learning and clarify how FML differs from existing work.
>
> ***Manifold-based Federated Learning.*** Several recent studies have explored manifold modeling to address client data heterogeneity in federated learning. A representative line of work uses manifolds as regularizers or calibration tools in feature space. For example, FedMRUR [1] introduces a hyperbolic graph manifold regularizer to reduce local-global inconsistency, FedMR [2] reshapes the feature manifold under partially class-disjoint data, and FedMC [3] models nonlinear local geometry and calibrates local features with a global geometry dictionary. Another related direction focuses on manifold alignment, such as FedMP [4], which performs stochastic feature manifold completion and aligns client manifolds with class prototypes. Different from these methods, FML treats manifolds as explicit and communication-efficient knowledge carriers: clients compress local class manifolds into representative points, the server adaptively fuses them through attention-based manifold mutual learning, and the resulting global manifold is fed back to guide local training.
>
> [1] Federated learning with manifold regularization and normalized update reaggregation. NeurIPS, 2023.
>
> [2] Federated learning under partially disjoint data via manifold reshaping[J]. Transactions MLR, 2023.
>
> [3] FedMC: Federated Manifold Calibration. ICLR, 2026.
>
> [4] FedMP: Tackling Medical Feature Heterogeneity in Federated Learning from a Manifold Perspective. arXiv, 2025.
>
> ## Regarding Weakness 2: Result Interpretation.
>
> Your valuable comments are crucial for helping us improve the clarity and rigor of our paper.
> First, the reported results are not from a single seed. As stated in Section 4.1 (Evaluation Metric) of the original manuscript, we repeat each experiment three times and use the average accuracy over the last five communication rounds as the final result to reduce randomness and improve stability. We agree that explicitly reporting the variance would provide a stronger indicator of robustness, and we will revise the final manuscript to include corresponding statistics (e.g., mean ± standard deviation).
>
> Second, as you pointed out, the improvement of FML over prior methods is smaller on Digits than on Office31. At the same time, it can be observed that all methods achieve higher accuracy on Digits than on Office31. This is because Office31 is a more challenging benchmark with stronger domain heterogeneity. Under this harder setting, FML yields more pronounced gains, further highlighting its effectiveness in handling severe cross-domain heterogeneity.
>
> ## Regarding Weakness 3: Scalability with a large number of clients.
> This is an excellent point, as scalability is key in FL. Although our original experiments featured a smaller number of clients, to address your concern, we have conducted new experiments on Digits, scaling up to 36 and 72 clients. The results demonstrate that the performance improvements from FML remain robust, proving its excellent scalability. We will add a table with these new results to our final manuscript.
>
> Table 1. Number of Clients M Impact on Performance (%).
> | Methods | M=18 | M=36 | M=72 |
> |---|---:|---:|---:|
> | FedAvg | 79.44 | 81.92 | 82.78 |
> | FedDIM | 81.24 | 82.62 | 85.71 |
> | FedDP | 80.77 | 82.13 | 84.22 |
> | FML | 82.91 | 85.81 | 88.98 |
>
> ## Regarding Key Questions: Local Epochs Robustness.
>
> Thank you for this insightful comment. It is highly valuable for clarifying the robustness of our method when communication becomes less frequent.  Following your suggestion, we have conducted additional ablation experiments on the Digits benchmark by increasing the number of local training epochs from the default 10 to 15 and 20. The results show that FML remains stable as the number of local updates increases, with only a very small change in average accuracy. We will add these results in our final manuscript. We attribute this robustness to the higher-quality calibrated samples produced by FML, which provide more reliable guidance for local training and lead to more stable local updates. We will include these results and discussion in the final manuscript.
>
> Table 2. Impact of local training epochs on performance (%).
> | Methods | 10 | 15 | 20 |
> |---|---:|---:|---:|
> | FedAvg | 79.44 | 77.87 | 76.61 |
> | FML | 82.91 | 82.76 | 82.57 |

---

> > ### Author Rebuttal · Reviewer_WcQ8 · 2026-04-04
> >
> > Thank you for addressing my concerns.

---

### Official Review · Reviewer_gvof · 2026-03-12

**Soundness:** 4
**Presentation:** 3
**Significance:** 4
**Originality:** 3
**Overall Recommendation:** 5
**Confidence:** 5

**Summary:**

This paper tackles a critical and challenging problem in Federated Learning (FL): domain heterogeneity. The authors insightfully point out the fundamental limitation of existing prototype-based methods—compressing a complex class distribution into a single point discards crucial structural information and intra-class variance across different domains. To overcome this, the authors propose a highly novel paradigm, Federated Manifold Learning (FML). The framework elegantly integrates Perceptual Manifold Compression (PMC) to reduce communication costs, an attention-based Manifold Mutual Learning (MML) mechanism for server-side fusion, and tailored manifold-guided loss functions (L_MA And L_MS) for local training. The theoretical motivation is sound, the methodology is comprehensive, and the empirical results on Digits and Office31 benchmarks demonstrate a significant performance leap (up to 6.48% improvement) over state-of-the-art methods. This is a solid, well-written, and impactful contribution to the FL community.

**Compliance With Llm Reviewing Policy:**

Affirmed.

**Final Justification:**

Thank you for your response, which has satisfactorily resolved my concerns. After reading the other reviewers' comments and the authors' responses, I have no further questions. Therefore, I am willing to increase my score to 5.

**Key Questions For Authors:**

Please see the weaknesses. If the author can address my concerns, I will reconsider the rating.

**Limitations:**

I suggest that the author add a discussion of limitations and future work to the Conclusion.

**Strengths And Weaknesses:**

Strengths:

1. The transition from "point-based" prototypes to "structure-based" perceptual manifolds is a brilliant insight. It naturally and effectively addresses the loss of domain-specific intra-class variance in heterogeneous FL scenarios.

2. Transmitting manifolds could ideally incur huge communication overhead. The authors cleverly employ the Determinantal Point Process (DPP) and introduce the Information Efficiency Index (IEI) to find the optimal trade-off between compression and information preservation, effectively solving the communication bottleneck.

3. Instead of naive averaging, the MML mechanism uses FINCH clustering and cross-attention to selectively fuse manifolds. This design ensures that domain-specific features are enriched rather than washed out, which is highly innovative.

Weaknesses:

1. I suggest the authors add a discussion regarding the distribution differences of optimal hyperparameters across different domains (e.g., Digits vs. Office31). Analyzing whether there is a significant shift in the optimal values of γ,α, and β would help readers better understand the robustness mechanisms of the framework across diverse scenarios.

2. While the paper successfully demonstrates the effectiveness of FML through quantitative results, adding a qualitative visualization in the appendix or the main text (e.g., comparing the distribution of manifolds in the feature space across different domains) would provide an intuitive understanding of how the framework captures cross-domain knowledge at a structural level.

3. It would be beneficial to add a brief discussion on practical deployment strategies concerning the trade-off between communication overhead and model accuracy. For instance, suggesting how to dynamically adjust the manifold size r based on the bandwidth of edge devices would further enhance the practical relevance of the study.

---

> ### Author Rebuttal · Authors · 2026-03-31
>
> Dear Reviewer gvof,
>
> We sincerely thank you for your thoughtful and constructive review. We are grateful for your positive evaluation and for recognizing the novelty of our Federated Manifold Learning (FML) framework, including the structure-based perceptual manifold design, the efficient compression strategy, and the manifold mutual learning (MML) mechanism.
>
> Your suggestions are highly valuable for improving the clarity and practical relevance of our work. We will incorporate all clarifications and supplementary results into the final version of the manuscript.
>
> ***
>
> ## Regarding Weakness 1: Hyperparameter Sensitivity Across Different Domains.
>
> We agree that analyzing hyperparameter trends is crucial for robustness. Following your suggestion, we conducted additional sensitivity experiments on the Digits benchmark for $\gamma, \alpha$, and $\beta$ (Tables 1-3). The optimal average performance on Digits is achieved at $\gamma=0.9, \alpha=0.7$, and $\beta=0.8$.
>
> Comparing this with Office-31 ($\gamma=0.7, \alpha=0.15, \beta=1.0$), we observe:
> 1. The optimal $\gamma$ is stable across benchmarks, suggesting cross-domain fusion is a robust component.
> 2. The optimal $\alpha$ is higher for Digits due to its larger domain gap (e.g., SVHN vs. MNIST), requiring stronger manifold alignment.
> 3. The optimal $\beta$ remains relatively close; the slightly larger value for Office-31 likely addresses its higher number of categories (31 vs. 10), requiring stronger inter-class separation.
>
> Table 1. Sensitivity analysis of hyperparameter $\gamma$ (%) on Digits.
> | $\gamma$ | 0.2 | 0.3 | 0.4 | 0.5 | 0.6 | 0.7 | 0.8 | 0.9 | 1.0 |
> | :--- | :---: | :---: | :---: | :---: | :---: | :---: | :---: | :---: | :---: |
> | Acc | 82.91 | 83.05 | 83.61 | 83.23 | 82.62 | 82.54 | 83.27 | 84.14 | 84.09 |
>
> Table 2. Sensitivity analysis of hyperparameter $\alpha$ (%) on Digits.
> | $\alpha$ | 0.05 | 0.1 | 0.15 | 0.2 | 0.3 | 0.5 | 0.7 | 0.9 |
> | :--- | :---: | :---: | :---: | :---: | :---: | :---: | :---: | :---: |
> | Acc | 84.06 | 82.59 | 82.54 | 83.32 | 82.41 | 83.46 | 84.25 | 83.90 |
>
> Table 3. Sensitivity analysis of hyperparameter $\beta$ (%) on Digits.
> | $\beta$ | 0.4 | 0.6 | 0.8 | 1.0 | 1.2 | 1.4 |
> | :--- | :---: | :---: | :---: | :---: | :---: | :---: |
> | Acc | 83.62 | 82.86 | 84.01 | 82.54 | 82.27 | 83.64 |
>
> ## Regarding Weakness 2: Qualitative Visualization of Cross-domain Manifold Structures.
>
> Following your suggestion, we conducted t-SNE visualizations on Digits using ResNet-18 features, provided in **Figure 1-4** via the anonymous link: [https://anonymous.4open.science/r/rebuttal-Anonymization/Anonymous_figure.pdf](https://anonymous.4open.science/r/rebuttal-Anonymization/Anonymous_figure.pdf).
>
> Comparing FedAvg and FML after 100 rounds:
> 1. **FedAvg**: The feature space remains disorganized, especially for SYN and SVHN, indicating that simple parameter averaging fails at cross-domain structural alignment.
> 2. **FML**: Formulates well-separated class clusters where samples from different domains within the same class exhibit a coherent manifold structure.
> These results provide intuitive evidence of FML's effectiveness in learning transferable manifolds.
>
> ## Regarding Weakness 3: Communication-Accuracy Trade-off in Practical Deployment.
>
> The manifold size $r$ serves as a control knob for communication efficiency. While $r \approx 10$ is our standard setting, $r$ can be adjusted dynamically based on client bandwidth.
> - Bandwidth-limited clients can upload smaller manifolds (e.g., $r=3$ or $r=5$).
> - High-connectivity clients can use larger $r$ (e.g., $r=10$) for richer structural info.
> Importantly, our MML module handles heterogeneous manifold sizes naturally as it operates on sets of points rather than requiring strict pointwise alignment, making FML highly suitable for real-world edge deployment.
>
> ## Regarding Limitations: Refining the Conclusion.
>
> We will revise the Conclusion to explicitly discuss current limitations and future directions, including:
> - Validation on larger-scale benchmarks and modern backbones.
> - Privacy-preserving extensions (e.g., DP or Secure Aggregation).
> - Adaptive communication strategies for heterogeneous edge environments.
>
> We thank you again for the constructive feedback, which has greatly inspired and improved our work.

---

> > ### Author Rebuttal · Reviewer_gvof · 2026-04-03
> >
> > Thank you for your response, which has satisfactorily resolved my concerns. After reading the other reviewers' comments and the authors' responses, I have no further questions. Therefore, I am willing to increase my score to 5.

---

### Official Review · Reviewer_Uzht · 2026-03-12

**Soundness:** 2
**Presentation:** 2
**Significance:** 2
**Originality:** 2
**Overall Recommendation:** 3
**Confidence:** 4

**Summary:**

This paper tackles domain heterogeneity in federated learning by arguing that existing prototype-based methods fail because they compress complex class distributions into single points, discarding crucial structural information. The authors propose Federated Manifold Learning (FML), which shifts to structure-based knowledge transfer by treating class features as perceptual manifolds—intrinsic geometric structures in feature space. Clients compress these manifolds via Determinantal Point Processes to preserve diversity while reducing communication, the server fuses them through an attention-based mutual learning mechanism, and local training is guided by manifold approximation and separation losses to align features with the global structure. Experiments on Digits and Office31 benchmarks show FML outperforms state-of-the-art methods by up to 6.48% in accuracy.

**Compliance With Llm Reviewing Policy:**

Affirmed.

**Final Justification:**

Thank you for your rebuttal and for addressing most of my concerns. I tend to raise my score to a weak reject.

**Key Questions For Authors:**

Please see weakness.

**Limitations:**

Yes.

**Strengths And Weaknesses:**

Strength:

1. This paper is overall well motivated.
2. There is novel concept breakthrough by proposing perceprtual manifold.

Weakness:

1. The manifold compression process using Determinantal Point Processes (DPP) presents a significant efficiency challenge. Specifically, when employing high-dimensional Vision Transformers (ViT), the computational complexity scales at least cubically O(n3). This creates a severe bottleneck for practical deployment.
2. The MML module relies on clustering (FINCH) and attention mechanisms. Consequently, the server-side aggregation complexity grows linearly or even super-linearly with the number of clients. This poses a critical scalability issue for industrial-scale Federated Learning systems, which often involve thousands of participating clients.
3. The paper lacks a rigorous security analysis. Transmitting feature vectors (even compressed manifold points) can be more susceptible to Feature Inversion Attacks compared to transmitting gradients or model parameters, potentially allowing adversaries to reconstruct approximate raw images. The manuscript does not discuss or integrate essential privacy-preserving mechanisms, such as Differential Privacy (DP) or Secure Multi-Party Computation (MPC).
4. The evaluation relies on traditional, small-scale, low-resolution datasets (e.g., Digits, MNIST, SVHN, Office31). The work lacks validation on large-scale benchmarks like ImageNet and does not utilize modern mainstream architectures (e.g., ResNet-50, ViT, or CLIP backbones). This limits the assessment of the method's generalizability and State-of-the-Art (SOTA) performance.

---

> ### Author Rebuttal · Authors · 2026-03-31
>
> Dear Reviewer Uzht,
>
> We sincerely thank you for your thoughtful and constructive review. We are greatly encouraged that you recognize our core contribution—knowledge transfer via perceptual manifold learning in the FML framework.
>
> Your insightful comments are valuable for improving the clarity, rigor, and completeness of our manuscript. We have carefully addressed all your concerns with detailed responses below and will include all revisions in the final paper.
>
> ***
>
> ## Regarding Weakness 1: Computational Overhead of DPP-based Manifold Compression
>
> We clarify that our work utilizes **Fast Greedy DPP** with a complexity of $\mathcal{O}(m^2 d + r^2 m)$, which is significantly more efficient than the $\mathcal{O}(m^3)$ complexity of standard DPP. Since $r \approx 10$ is much smaller than $m$ in our setting, the selection overhead remains negligible. Our implementation (verified in our code) is provided below:
>
> ```python
> def dpp(feature_vectors: np.ndarray, score: np.ndarray = None, max_length: int = None, epsilon: float = 1e-10):
>
>     item_size = feature_vectors.shape[0]
>     kernel_matrix = np.dot(feature_vectors, feature_vectors.T)
>     cis = np.zeros((max_length, item_size))
>     di2s = np.copy(np.diag(kernel_matrix))
>     selected_items = list()
>     selected_item = np.argmax(di2s)
>     selected_items.append(selected_item)
>
>     while len(selected_items) < max_length:
>         k = len(selected_items) - 1
>         ci_optimal = cis[:k, selected_item]
>         di_optimal = math.sqrt(di2s[selected_item])
>         elements = kernel_matrix[selected_item, :]
>         eis = (elements - np.dot(ci_optimal, cis[:k, :])) / di_optimal
>         cis[k, :] = eis
>         di2s -= np.square(eis)
>         selected_item = np.argmax(di2s)
>         if di2s[selected_item] < epsilon:
>             break
>         selected_items.append(selected_item)
>     return selected_items
> ```
> ## Regarding Weakness 2: Computational Scalability of Clustering and Attention in MML.
>
> Thank you for your rigorous comment. We clarify that the server-side MML module operates on highly compressed manifold points instead of raw features. With $r=10$ representative points per class, the input size for FINCH clustering is only $M \times r$, which is computationally lightweight.
>
> FINCH is efficient and the cluster count depends on the real feature structure rather than the client number. Even with $1000$ clients, it often produces very few clusters. The subsequent self-attention is only applied across clusters, so server computation remains practical.
>
> ## Regarding Weakness 3: Privacy Risks of Feature-based Knowledge Sharing.
>
> Thank you for your valuable suggestion. We agree that sharing class-level manifold representations may introduce privacy risks, since they are constructed from representative local features and therefore do not provide a formal guarantee against feature inversion or related attacks.
>
> The main focus of this work is to improve knowledge transfer under heterogeneous federated learning rather than to design a privacy-preserving mechanism. We will revise the manuscript to explicitly discuss this limitation and note that integrating privacy-preserving techniques such as DP, secure aggregation, or MPC is an important direction for future work.
>
> ## Regarding Weakness 4: Evaluation on Large-scale Datasets and Modern Backbones.
>
> Your suggestion on validating with larger models and datasets is valuable. We clarify that our original experiments use Digits, Office, and ResNet-18 to follow standard evaluation protocols in heterogeneous federated learning, ensuring fair comparison with mainstream methods:
>
> - [1] Geometric knowledge-guided localized global distribution alignment for federated learning. CVPR, 2025.
> - [2] Re-fed+: A better replay strategy for federated incremental learning. IEEE TPAMI, 2025.
> - [3] Federated domain generalization with decision insight matrix. IJCAI, 2025.
>
> Meanwhile, to better validate generalizability and scalability, we further conduct experiments on the larger Office-Home dataset and the stronger ResNet-50 backbone. Results confirm our method remains effective in more challenging settings.
>
> Table 1. Comparison with the SOTA methods on Office-Home dataset.
>
> | Methods | Art | Clipart | Product | Real World | AVG |
> | :--- | :---: | :---: | :---: | :---: | :---: |
> | FedAvg | 55.18 | 46.27 | 73.51 | 66.28 | 60.31 |
> | FedProto | 54.42 | 46.92 | 72.61 | 64.85 | 59.70 |
> | FML (Ours) | 62.12 | 58.39 | 77.29 | 76.27 | 68.52 |
>
> We have added supplementary experiments (detailed in our global response/anonymous PDF) including:
> - **Hyperparameter sensitivity**: Analysis of $\gamma, \alpha$, and $\beta$ on the Digits dataset.
> - **Visualizations**: t-SNE manifold distributions across different domains.
> - **Scalability**: Performance under increased clients and local training epochs.
> - **Efficiency**: Quantification of communication overhead.
>
> We thank you again for the constructive feedback, which has greatly improved our work.

---

### Decision · Program_Chairs · 2026-04-30

**Decision:**

Accept (regular)

**Comment:**

There are two positive reviewers and two negative reviewers. This work's strengths include clear description about motivation, novel concepts in method design, many baselines in experiment comparison, good writing. The common concern from the negative reviewers about validation on larger-scale data has been addressed in rebuttal with supportive results. Overall, this paper is a decent submission for weak acceptance.